# MULTICHARME: A modified Chernin-type multi-pass cell designed for IR and THz long-path absorption measurements in the CHARME atmospheric simulation chamber

Jean Decker[1], Éric Fertein[1], Jonas Bruckhuisen[1], Nicolas Houzel[1], Pierre Kulinski[1], Bo Fang[2], Weixiong Zhao[2], Francis Hindle[1], Guillaume Dhont[1], Robin Bocquet[1], Gaël Mouret[1], Cécile Coeur[1] and Arnaud Cuisset[1]

[1]Laboratoire de Physico-Chimie de l'Atmosphère, UR4493, LPCA, Université du Littoral Côte d'Opale, F-59140 Dunkerque, France.
[2]Laboratory of Atmospheric Physico-Chemistry, Anhui Institute of Optics and Fine Mechanics, HFIPS, Chinese Academy of Sciences, Hefei 230031, Anhui, China

*Correspondence to*: Arnaud Cuisset (arnaud.cuisset@univ-littoral.fr)

## Abstract

We have developed MULTICHARME, a modified Chernin-type multi-pass cell especially designed for IR and THz long-path absorption measurements in the CHamber for Atmospheric Reactivity and Metrology of the Environment (CHARME). By measuring the output power using a near-IR diode-laser and a THz amplified multiplication chain, we have established that the effective reflectivity of MULTICHARME is better than 94% over approximately three decades of frequency. Absorption measurements of $N_2O$ have been performed by probing highly excited rovibrational transitions in the near-IR and ground state rotational transitions at submillimetre wavelengths. In each case the linearity of the absorbance with the pathlengths was verified. Finally, we demonstrate that THz spectroscopy is able to study the isotopic composition of greenhouse polar gases such as $N_2O$ and to absolutely quantify stable ($N_2O$) and reactive ($O_3$) species at trace levels. At low pressure the ozone concentration was continuously monitored and its decay characterised. The deduced ozone lifetime of $3.4 \pm 0.1$ h is shorter compared with previous measurements performed in CHARME at atmospheric pressure. For the first time, the ability of THz rotational spectroscopy to monitor, with a very high degree of selectivity, stable and reactive polar compounds at trace level in an atmospheric simulation chamber is demonstrated. However, the sensitivity of the THz monitoring needs to be improved to reach atmospheric trace levels. For this purpose, it is necessary to fully understand the origin of the observed baseline variations caused by the complex multiple standing waves present in MULTICHARME.

## 1 Introduction

Atmospheric Simulation Chambers (ASC) have been developed to study atmospheric physico-chemical processes under controlled conditions. Aside small Teflon bags of few hundred litres, approximately 35 chambers are currently in operation around the world. All these reactors are equipped with a large variety of instruments dedicated to the monitoring of

gases and particles, ranging from commercially available apparati to specialized custom-designed setups offering in-situ measurement of chemical species by optical techniques.

Various spectrometers based on photonic sources from UV to mid-IR have been coupled to different ASCs allowing the stable and unstable reactants to be monitored along with the gas and particle phase products involved in key atmospheric reactions (Barnes et al., 1994; Bloss et al., 2005b; Rohrer et al., 2005; Wagner et al., 2006; Ren et al., 2017; Massabò et al., 2018). The identification of critical intermediate species and the time-resolved quantification of kinetic parameters is of upmost importance for atmospheric models . The optical components used to couple the probe beam with the ASC are selected depending on the nature of the light source employed, the detection scheme and the geometry of the reactor. The trace gas monitoring requirement in an ASC makes it necessary to reach sub-ppm limits of detection (LOD) by maximizing the interaction pathlength between the light and the probed species. Nowadays, most of the ASCs, using optical spectrometers as analysers, are equipped with an optical system enabling multiple passes of the probe light back and forth across the ASC, thus increasing the absorption pathlength and consequently improving the LOD.

For example, several ASCs are coupled to commercially available Fourier-Transform infrared (FTIR) spectrometers for Volatile Organic Compound (VOC) detection at low-resolution ($>0.5$ cm$^{-1}$) in the mid-IR (typically 400 - 4000 cm$^{-1}$) using a White-cell multi-pass mirror configuration reaching several hundreds of meters of interaction pathlengths with a broadband black-body IR source (White, 1942). The following ASCs are well known examples in the EUROCHAMP 2020 network (EUROCHAMP 2020): (i) the 4.2 m$^3$ CESAM chamber in Paris uses a globar source inside a Bruker interferometer with a 192 m pathlength White-cell (Wang et al., 2011); (ii) the quartz reactor QUAREC in Wuppertal also employs a White-cell configuration to achieve an optical pathlength of $484.7 \pm 0.8$ m and is entirely  mounted inside the photoreactor for sensitive in situ long path IR absorption monitoring of reactants and products (Barnes et al., 1994); (iii) the EUPHORE photoreactor in Valencia makes it possible to intercompare 1 cm$^{-1}$ resolution FTIR data (L=616 m) with Differential Optical Absorption Spectroscopy measurements in the 389-469 nm UV range with a dedicated White-cell (W-DOAS) which can reach km pathlengths (Bloss et al., 2005a, b).

The HIRAC chamber in Leeds is coupled to a LIF-FAGE analyzer and to a multiple pass FTIR system. In contrast to the previous examples of traditional White-type arrangements, the HIRAC team chose a modified multiple pass matrix system developed by Chernin (Chernin and Barskaya, 1991; Chernin, 2001). This solution retains the focal properties of the original White cell and perfectly conserves optical throughput over a range of matrix arrangements. In practice, the Chernin-cell is very easy to align, and shows very good stability to vibrations, with the FTIR giving good LODs over short acquisition times. For observation times as short as 1 min, LODs below 100 ppbv are obtained for ozone and VOCs such as acetaldehyde, methane, and formaldehyde with the FTIR interferometer coupled to the 128.5 m Chernin cell (Glowacki et al., 2007b). Finally, the Chernin multi-pass cell optimizes the recirculation of the beam over many focused lines on the field mirrors and minimizes the overlapping between adjacent refocusing points, thus facilitating the control the propagation of more divergent beams over long distances.

Due to the lack of reliable and sufficiently powerful sources and due to the difficulty to control on long distances the propagation of more divergent beams, far-IR/Terahertz (THz) spectrometers have never been used for trace gas monitoring in ASCs. Yet, in 2013, Kwabia Tchana et al. have demonstrated the ability to perform FT-far-IR spectroscopy in a large cryogenically cooled Chernin-cell (Kwabia Tchana et al., 2013). This cell is coupled to the AILES beamline of the SOLEIL synchrotron (Brubach et al., 2010) and allows mid- and far-IR measurements with variable pathlengths from 3 to more than 141 m thanks to exceptional properties of brightness and small divergence especially in the THz domain.

In this study, we present and characterize a multiple pass system developed for the CHamber for Atmospheric Reactivity and Metrology of the Environment (CHARME) (Fayad, 2019). Based on a Chernin type arrangement, the so named MULTICHARME has been dimensioned for the CHARME ASC and allows the monitoring of stable and reactive atmospheric species at trace levels over three decades of frequencies by probing long path rovibrational and rotational molecular absorbances respectively in the IR and in the THz domains. To the best of our knowledge, this is the first time that a THz spectrometer is used for in situ measurements of atmospheric species in an ASC. The results obtained in this study highlight a new approach based on pure rotational spectroscopy for rapid and highly selective detection of stable and reactive atmospheric compounds in a simulated atmospheric environment. The first part of the article will be dedicated to the description of CHARME, MULTICHARME and its coupling with IR and THz sources. Results are presented and discussed in the second part of the article with a special focus on the THz measurements which are unprecedented in the ASC community.

## 2 Experimental setup and Methodology

### 2.1 CHARME (CHamber for Atmospheric Reactivity and Metrology of the Environment)

CHARME is the new atmospheric simulation chamber designed in the LPCA (Laboratoire de Physico-Chimie de l'Atmosphère) laboratory in Dunkirk (France). CHARME is described in details in Fayad et al. 2019. Briefly, it consists of a 9.2 $m^3$ evacuable cylinder (length $\approx$ 4 m; internal diameter $\approx$ 1.7 m) made of electropolished stainless steel (304 L). The inner surface of the reactor is mechanically polished in addition to an electrochemical treatment, which enhances the light reflectivity and also reduces the interaction of gases and particles with the walls. Four stainless steel fans (diameter 50 cm) located at the bottom of the chamber assure a fast homogenization of the reactive mixtures and can be activated only at atmospheric pressure. The rigid walls with a thickness varying from 4 to 40 mm permit the reactor to guarantee vacuum and non-deformation of the flanges. Consequently, CHARME is considered as vacuum compatible, and it is capable to support most of the mechanical constraints when it is under vacuum. The range of pressure under which it can operate is: 0.05 mbar < P < 1000 mbar. CHARME is pumped with a vacuum pump (Cobra NC0100-0300B) and is filled with purified and dried air at the required pressure using a generator (Parker Zander KA-MT 1-8). The pressure within the chamber is measured using 2 MKS BARATRON (626B13MDE (1000 mbar) and 626B01MDE (1 mbar)) and a pressure reader (MKS PR4000B) and the relative humidity and temperature are monitored by a combined probe (Vaisala HUMICAP, HMT330).

CHARME has 9 different circular access ports ranging from 45 to 20 cm in diameter (see Fig. S1 in the Supplement), which are used for various tasks: accommodate the MULTICHARME optical setup for a Chernin-cell described in this article; to provide physical access to the inside of the chamber for any cleaning and alignment operations; for in-situ monitoring of gases and particles by IBBCEAS (Incoherent Broad Band Cavity Enhanced Absorption Spectroscopy) (Fiedler et al., 2003; Meng et al., 2020); to introduce gases and/or particles into the chamber.

## 2.2 MULTICHARME (MULTI-pass cell specially designed for CHARME)

### 2.2.1 General design

In order to ensure a sufficient optical throughput at THz frequencies the MULTICHARME Chernin-cell configuration employed uses two rectangular field mirrors (266 × 310 mm and 222 × 50 mm), and three circular objective mirrors (diameter 130 mm). We have opted for the modified version of the Chernin cell with the input and output windows on both sides of the field mirrors (Chernin, 2001). MULTICHARME has been designed in order to maximize the size of the mirrors to account for the large size of the beams and the strong divergences at longer wavelengths in the THz domain. We were nevertheless limited by the size of the flanges coupled to the DN 450 circular ports of the CHARME ASC. All mirrors have the same radii of curvature (ROC = 5000 mm) corresponding to the MULTICHARME base-length. Fused silica substrate were used for the field mirrors, K9 for the objectives. To optimize the reflectivity from near-IR to THz domain i.e. three frequency decades from 300 THz to 0.3 THz, a coating of 500 nm silver protected by 10 nm of $Al_2O_3$ was used. These relatively large mirrors were manufactured by the Anhui Institute of Optical and Fine Mechanics over a period of several months to build the substrates and to deposit the coatings. This institute has already developed with success a Chernin cell for the detection of atmospheric radicals with Faraday rotation spectroscopy in the mid-IR (Fang et al., 2019).

The two optical mounts, equipped with the three objective mirrors and the two field mirrors, constitute the Chernin cell presented in Fig. 1. Using the DN 450 access ports (A1 and A2 in Figure S1 in the Supplement) and custom manufactured vacuum enclosures, they are located on opposite ends and placing the mirrors inside of the cylindrical chamber. Two aluminum optical mounts were constructed to hold the mirrors. The mass of the additional components added to CHARME was 160 kg for each of the two DN 450 ports located at the opposite ends.

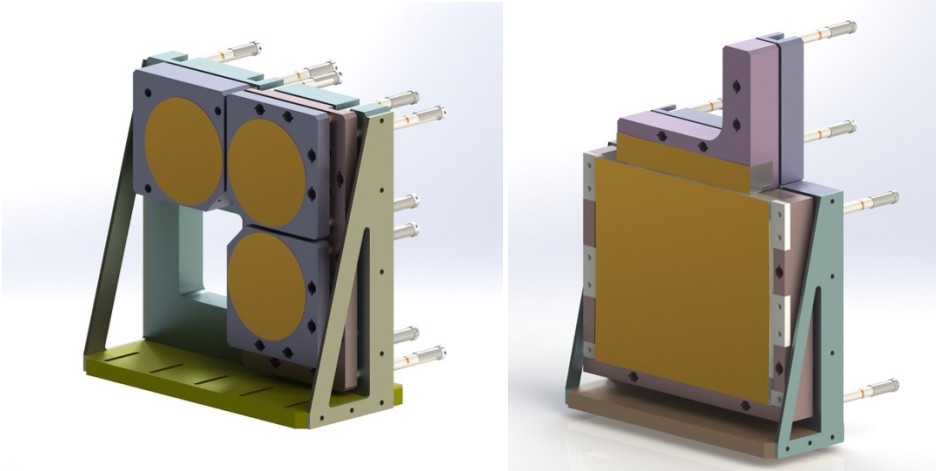

**Figure 1.** Schemes of the two mirror assemblies of MULTICHARME: the objective block (left) and the field block (right)

The total deformation of the Chernin cell and the corresponding mounting flanges was analyzed under static conditions. The forces that considered were their own gravity and the atmospheric pressure on the outer surface of the flanges when the chamber was in a vacuum state. The maximal displacements have been estimated to be less than 30 μm. A picture of the five mirrors is given in Fig. S2 in the Supplement. The characterization of their reflectivity in the IR and in the THz domains is described in Sect. 2.2.2 and 2.2.3, respectively. In order to accurately and easily control the optical alignment and the pathlength, both the field and objective mirrors are equipped with computer-controlled micrometric screws: MULTICHARME may be aligned with 18 compact linear motorized actuators TRA25PP from the MKS/Newport company (six actuators are fixed on the field mount to adjust the three degrees of freedom of the two field mirrors; twelve actuators are on the objective mount: nine to control the position of the three objective mirrors and three to adjust the relative orientation between field and objective blocks). The actuators are controlled by a home-made ARDUINO based system located outside the cell (see Fig. S3 in the Supplement). This system allows to select independently each actuator and to control their movement. The '5-mirror configuration', easy-to-align, forms a matrix with an adjustable number of rows and even columns on the field mirrors, and offers good vibration stability. Different spot patterns obtained with a He-Ne alignment laser are shown in Fig. S4 in the Supplement highlighting our ability to adjust matrix arrangement on the field mirror for different pathlengths in MULTICHARME from 120 m to 540 m.

**2.2.2 Coupling with IR tunable diode laser**

The characterization in the near-IR has been performed using a continuous-wave External Cavity Diode Laser (ECDL, Toptica DL pro) tunable from 1340 to 1450 nm with an output power of 80 mW, a spectral width of 100 kHz and a mode-hop free tuning range of 20 GHz. The ECDL source, the photodetector and the transfer optics were placed on an optical breadboard

fixed to the field flange. The IR input and output was coupled to the CHARME ASC by two 3 inch diameter ZnSe windows with a 1° wedge giving a theoretical transmission of about 70% at 1.4 µm.

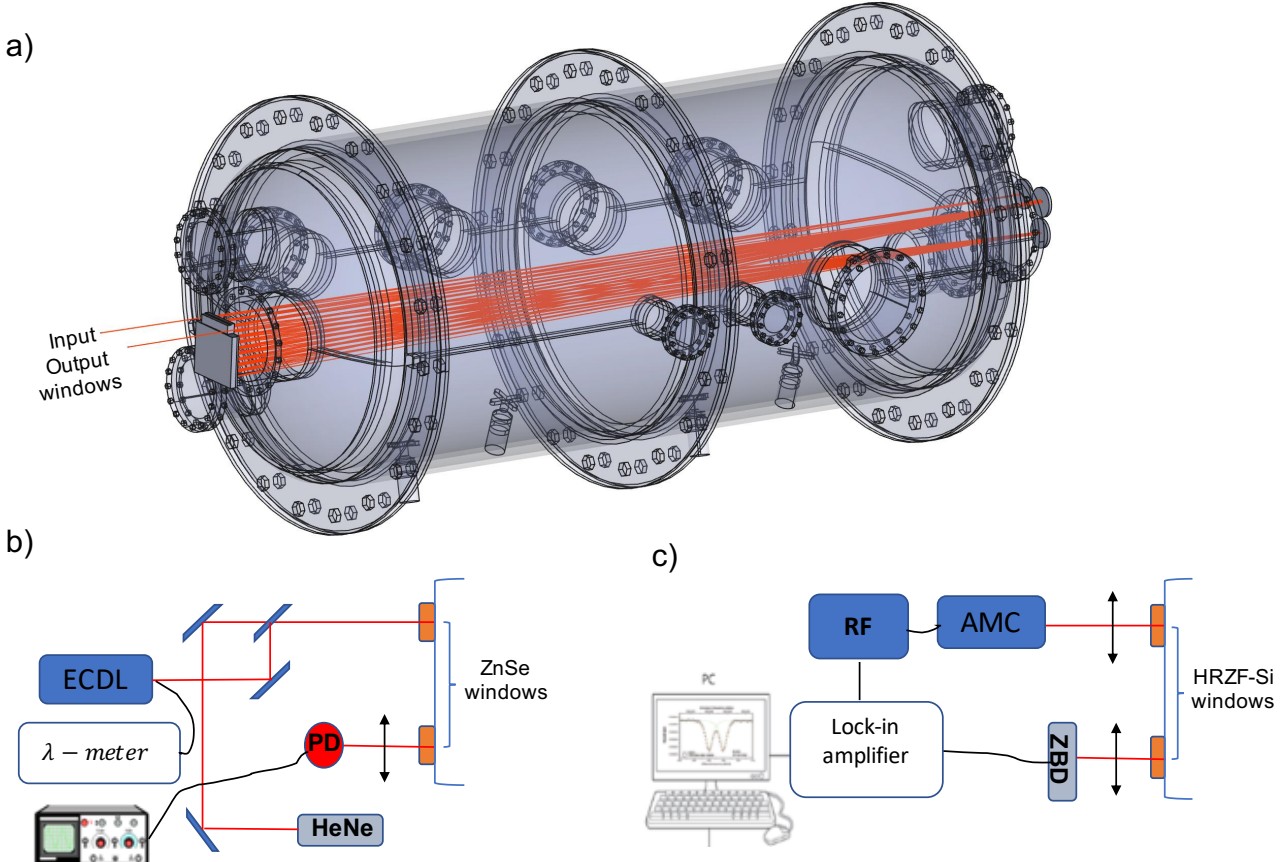

**Figure 2.** (a) 3D plot of MULTICHARME in CHARME performed with the FreeCad software and the Optics Workbench. (b) near-IR coupling to MULTICHARME via ZnSe windows using: an External Cavity Diode Laser (ECDL); a standard photodiode (PD); a He-Ne laser for alignment and a wavelength meter ($\lambda - meter$). (c) THz coupling to MULTICHARME via High Resistivity Float Zone -Silicon (HRFZ-Si) windows using: an amplified multiplication chain (AMC) driven by a RF synthesizer; a zero-biased detector (ZBD) connected to a lock-in amplifier.

The IR optical configuration is presented in Fig. 2a, an additional mirror was inserted into the beam path to coaxially inject a red MKS/Newport He-Ne Laser (5 mW) with the IR beam axis in order to facilitate the alignment of the Chernin-cell. This enables the IR pathlength in MULTICHARME to be evaluated by the observation of the matrix arrangement of the He-Ne spots on the field mirrors observed from a BK7 window placed on the objectives transfer flange. All adjustments and pathlength changes can be made *in situ,* without venting CHARME, with the computer-controlled actuators. Once the laser was adjusted for the desired operating range, a wavelength calibration was performed (Burleigh WA-1500) with an accuracy

better than 4 x 10^-3 cm^-1. An InGaAs detector (Thorlabs PDA400) with a typical bandwidth of 10MHz was used for the detection. Spectra were obtained by applying a voltage ramp to the piezo actuator allowing an excursion of 0.17 cm^-1 around the line-center at a frequency of 1.3 Hz. The received photodiode signal was averaged by a digital oscilloscope (DSO-X 2002A Agilent Technologies, maximum frequency of 70 MHz). The signal was typically accumulated over 16 ramp cycles with a sampling of 12.5 kHz (10 bits of vertical resolution).

To characterize the performance of MULTICHARME in the near-IR, we have examined the optical throughput in the modified Chernin multi-pass cell by measuring the output IR power for several different matrix arrangements(Glowacki et al., 2007a). It corresponds to 12 matrix configurations from 24 to 108 passes, for pathlengths from 120 m (3 rows × 4 columns) to 540 m (9 rows × 6 columns), respectively. The output powers shown in Fig. 3 were measured with a PDA400 power-meter from the Thorlabs company with an accuracy of 1.2 μW. The variation of the output power with the number of reflections was modelled by a power law $P = A * R_{eff}^{n}$, where $A$ is a constant corresponding to the received power with no reflection, $R_{eff}$ is the effective mirror reflectivity at 1.4 μm and $n$ is the number of reflections. An effective reflectivity of the mirrors at 1.4 μm was adjusted to 95.98 ± 0.06% (error given by the fit) with a weighted fit using the estimated error bars taking into account the accuracy of the IR photodetector. This value is rather good considering the expected decrease of the IR reflectance at shorter wavelengths.

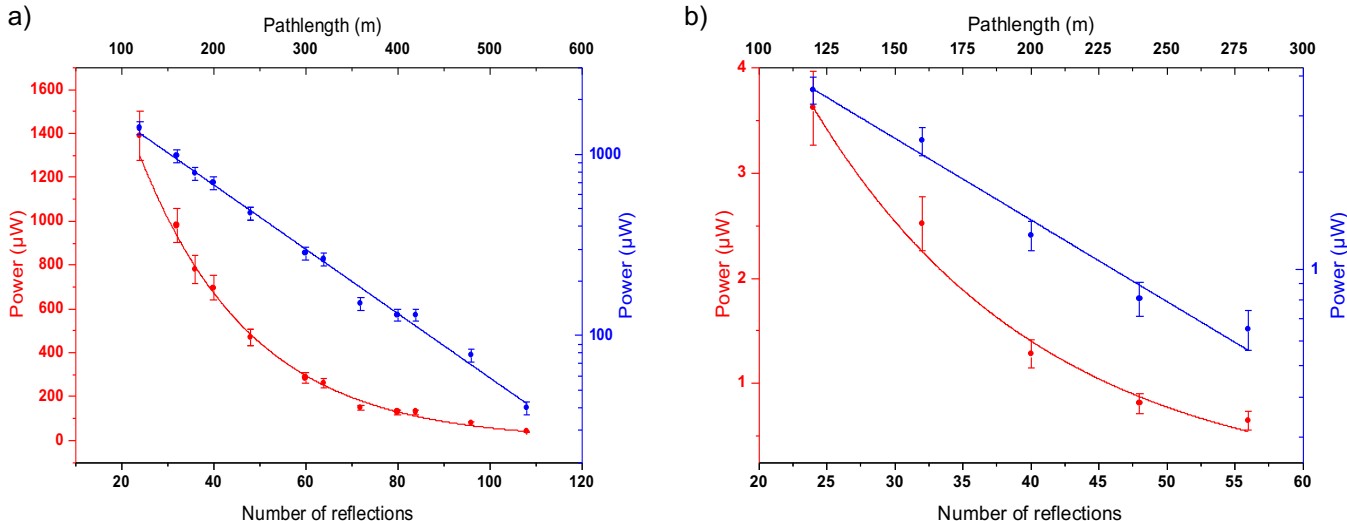

**Figure 3.** Attenuation of the IR (a) and the THz (b) output power (linear scale in red, log. scale in blue) vs the number of the reflections in MULTICHARME. The effective IR and THz reflectivities are deduced from the fit of the measurements with the power law: $P = A * R_{eff}^{n}$ (power law fit in red, linear fit in blue). The fits are performed with an instrumental weighting $\omega_i = 1/{\sigma_i}^2$ with $\sigma_i$ being the error bar sizes taking into account: the IR sensor uncertainty (5%) and the accuracy of the controller sensor (3%) for (a); the THz power fluctuations measured on the lock-in amplifier for (b). See Fig. S4 in the Supplement for spot's matrix illustrations.

### 2.2.3 Coupling with THz source

Several compact and versatile solid-state sub-THz sources are currently used in our laboratory for trace gas monitoring using high resolution rotational signatures of atmospheric pollutants in realistic media, e.g. industrial combustion or foodstuffs packed gas sample (Mouret et al., 2013; Hindle et al., 2018). In this study, a commercial Virginia Diode Inc amplified multiplication chain (AMC) driven by a microwave synthesizer (Rhodes & Schwarz SMA 100B) was used. Two Menlo systems TPX lenses of 2" diameter and 100 mm focal length were used to collimate the beam at the entrance of the chamber and refocus the output beam onto the detector. The propagation of the THz radiation in MULTICHARME was modelled as a Gaussian beam using a program developed by Lightmachinery inc. (Lightmachinery) showing that the use of f =100 mm lenses provides the best results taking into account the beam properties (waist and divergence), the dimension of MULTICHARME and the different losses (transmission through the windows & reflections on the field and objective mirrors). Two different AMC multiplication stages were used, the first (multiplication ×18) to cover the 140 to 220 GHz frequency region with an average power of 3 mW and the second (multiplication ×54) to cover the 440 to 660 GHz region with an output power of around 50 µW. To detect the MULTICHARME output THz signal, two VDI Schottky Zero Biased Detectors (ZBD) WR5.1 and WR1.5 were chosen with typical responsivities of 1000 V/W and 2400 V/W, respectively for the first and the second stages. The typical Noise Equivalent Power (NEP) of these Schottky diodes is estimated to 10 pW/$\sqrt{Hz}$. Both Amplitude Modulation (AM) or Frequency Modulation (FM) schemes were employed. The simultaneous use of AM and FM as a function of frequency proved useful to minimise the effects of the standing waves in the baseline, (see Sect. 3). A computer controlled Ametek 7230 lock-in amplifier recovered the measured signal as a function of the frequency. High resistivity float zone silicon windows with a theoretical transmission of more than 50% were used for the entry and exit of MULTICHARME during the THz measurements. This material is opaque at the He-Ne laser wavelength, so the THz beam alignment in MULTICHARME was performed by superimposing the THz beam to the IR beam at the entrance of MULTICHARME and on the objective mirrors.

As we did with the IR radiation, we have measured the THz output power for different pathlengths. These measurements presented in Fig. 3b were performed around 190 GHz corresponding to the maximal output power at the rank ×18 of the AMC. With a lower input power and a more divergent beam, only 5 matrix configurations corresponding to pathlengths from 120 m (24 passes) to 280 m (56 passes) were accessible at this frequency. If we consider just the detector NEP, 50 pW should be detectable with a time constant of 1 s, i.e. four orders of magnitude below the power level measured for a 280 m pathlength. Here the strong limitation to reach larger pathlengths is the divergence of the THz radiation and the size of the THz waist on the field mirrors. Reaching a pathlength of 280 m with an amplified frequency multiplication chain which is highly divergent source is a significant improvement compared commonly used setups. Extending the pathlength further should be possible for higher frequencies or with more powerful THz sources. An overview of the best performance reached by rotational submm-wave/THz long-path absorption spectroscopy is provided (Cuisset et al., 2021): the maximal THz pathlength in a White-cell was obtained with a weakly divergent and bright synchrotron source but it never exceeded 200 m (Brubach et al., 2010); longer

interaction pathlength are accessible only in Fabry-Perot resonators with intra-cavity spectroscopic techniques (Hindle et al., 2019). By fitting, with the power law, the THz output power, an effective THz reflectivity of $94.2 \pm 0.4$ % (error given by the fit) has been found for MULTICHARME. This value is only slightly lower than the near-IR value found in Fig. 2a. Therefore, MULTICHARME guarantees a reflectivity better than 94% on more than three decades of frequencies. With a unique modified Chernin type multi-pass cell, long path-absorption spectroscopic measurements are now possible from the THz to the near-IR domain in an ASC such as CHARME.

## 3 Results and discussion

### 3.1 Absorption linearities

The first spectroscopic measurements were carried out to verify the linearity of the absorption over three decades of frequencies from mm-wave to near-IR domains. In the aim to characterize the performances of MULTICHARME, we have chosen nitrous oxide $N_2O$ as test molecule for three main reasons: (i) $N_2O$ is a powerful and very stable greenhouse gas which can be considered now as the dominant ozone-depleting substance emitted in the 21st century in our atmosphere (Ravishankara et al., 2009). The monitoring of its chemical activity during its transport from the troposphere to the stratosphere is crucial to control the ozone depletion; (ii) $N_2O$ is actually monitored in the troposphere and in the stratosphere by probing its rovibrational IR and rotational THz transitions with sounders such as IASI (Clerbaux et al., 2009) or TALIS (Wang et al., 2020) respectively; (iii) The molecular rotational and rovibrational line parameters (line frequencies, line widths, line intensities) of $N_2O$ were measured and calculated from the mm-wave to the near-IR domains and are listed in the international spectroscopic databases such as JPL (Pickett et al., 1998) or HITRAN (Gordon et al., 2022).

The absorption linearity was first checked in the near-IR by probing the R(17) rovibrational transition of $N_2O$ in the highly excited $(3, 2, 0, 1) \leftarrow (0, 0, 0, 0)$ vibrational band. From the HITRAN database (Gordon et al., 2022), this line is predicted with a weak intensity (S=$7.1\times10^{-25}$ cm$^{-1}$/(molecule.cm$^{-2}$)) and is expected at 7149.45 cm$^{-1}$, where the ECDL performs optimally. As shown in Fig. 4a, the near IR rovibrational absorbance has been obtained for 8 different paths in MULTICHARME covering interaction distances from 120 m to 480 m at a pressure of 20 mbar. In order to avoid any saturation of the absorption signal, a calibrated mixture of $\chi$=1000 ppmv of $N_2O$ diluted in $N_2$ was used. In order to deduce the absorbance given by $A(\nu) = ln \frac{I_0(\nu)}{I(\nu)}$, baseline variations $I_0(\nu)$ were measured systematically with the signal variations $I(\nu)$. The wavenumber calibration was performed with the measurements of the WA-1500 wavelength-meter with an accuracy better than $4\times10^{-3}$ cm$^{-1}$. The integrated absorbances, converted in cm$^{-1}$ units, are determined by fitting the Doppler broadened near-IR lines with Gaussian profiles. They have been plotted in Fig. 4b according to the associated pathlengths. A linear regression weighted with the error bars ($3\sigma$) deduced from the Gaussian fits yields an R$^2$ of 0.998 and a slope s=1.17(7) $\times10^{-5}$ cm$^{-1}$/m. The linearity observed ensures the absence of saturation and guarantees that the IR photons introduced into MULTICHARME have traveled the same optical path before they reach the detector (Kwabia Tchana et al., 2013). From the measured slope in

Fig. 4b and the averaged linewidth estimated to be $\Delta\nu$ =0.014 cm$^{-1}$ (HWHM), we can deduce, according to the equation $\alpha_{0,exp} = \sqrt[2]{\frac{ln2}{\pi}}\frac{s}{\Delta\nu}$ (Sigrist, 1994), an experimental value of the maximal absorption coefficient of the line $\alpha_{0,exp} = 3.92 \times 10^{-6}$ cm$^{-1}$, around 50 times bigger than the expected value $\alpha_{0,th} = 7.26 \times 10^{-8}$ cm$^{-1}$ given by the equation $\alpha_{0,th} = \frac{S\chi}{k_B T \gamma_{air}}$ (Hindle et al., 2018), where the line intensity S and the air-broadening coefficient $\gamma_{air}$ are the tabulated values in the HITRAN database. The variation between $\alpha_{0,exp}$ deduced from our measurements and $\alpha_{0,th}$ could be partially explained by significant uncertainties of the tabulated near IR parameters S and $\gamma_{air}$ from a very weak rovibrational line which was, to the best of our knowledge, never measured before this study.

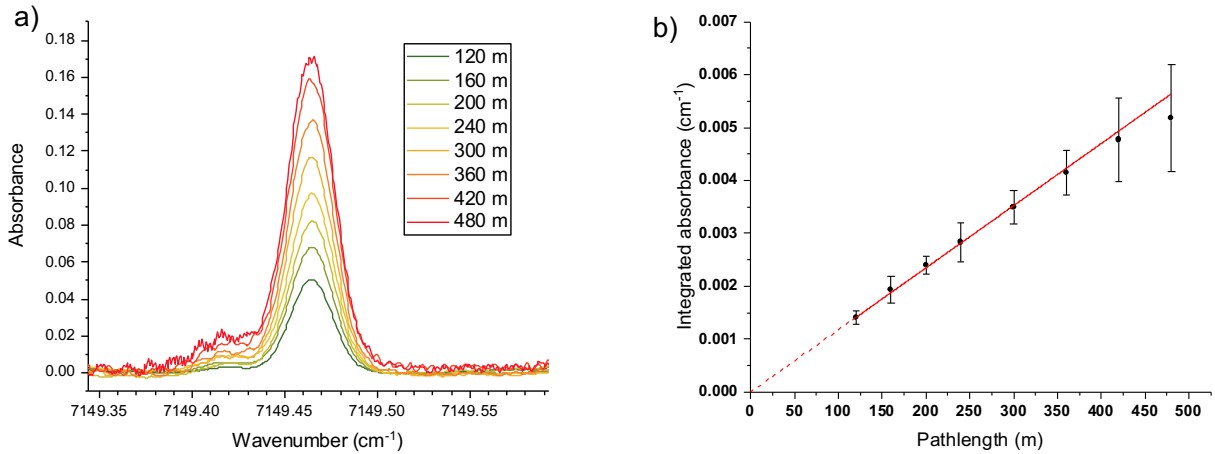

**Figure 4.** (a) Absorbance of the R(17) rovibrational line of N$_2$O in the $(3, 2, 0, 1) \leftarrow (0, 0, 0, 0)$ vibrational band measured with the ECDL IR source in the 7149.5 cm$^{-1}$ region for 8 different pathlengths in MULTICHARME ($\chi_{N2}$=1000 ppmv and P=20 mbar). The shoulder observed at low frequency is assigned to the ECDL source which is not perfectly monomode in this region. (b) Linear fit of pathlength dependence of the integrated absorbances, instrumentally weighted on the error bars deduced from the Gaussian fits of the absorbances shown in Fig. 4a.

The absorption linearity was also checked in the THz domain by probing the pure R(22) rotational transition of N$_2$O in its ground state expected at 577578.215 MHz with a line intensity estimated to S=2.9×10$^{-22}$ cm$^{-1}$/(molecule.cm$^{-2}$) from HITRAN database (Gordon et al., 2022), 400 times stronger than the previously probed near-IR rovibrational line and experimentally measured (Rohart et al., 2003). For the THz measurement, the mixing ratio in air was 400 ppmv and the rotational lines were measured at a total pressure of 4 mbar. Unlike IR rovibrational bands, with only few mbar of pressure, the collisional broadening is dominant compared to the Doppler broadening and a Lorentzian profile was assumed to fit these THz lines. The measured absorbances for five different pathlengths from 120 m to 280 m are presented in Fig. 5a. Compared to the IR lines, strong baseline oscillations due to standing THz waves affect the line profile especially for the longest THz

paths. In Fig. 5b, we have determined the integrated absorbances by determining the area of the rotational lines shown in 5.a. Compared to the IR results, larger uncertainties are deduced from the integration process due to the baseline variations. Nevertheless a linear behavior is estimated to be with the different pathlengths at least up to 240 m which constitutes already a record for a THz radiation in a multi-pass cell (Cuisset et al., 2021). With s = 5.1(4) × 10$^{-7}$ cm$^{-1}$/m, the slope obtained with the four THz measurements from 120 m to 240 m is smaller than the IR one. The THz rotational linewidth is estimated to be 3.4× 10$^{-4}$ cm$^{-1}$ (HWHM), around 300 times smaller than the IR rovibrational linewidth highlighting the excellent selectivity of the THz spectroscopy compared to the IR one due to a weaker Doppler broadening. An associated value of $\alpha_{0,exp} = 4.77 \times 10^{-6}$ cm$^{-1}$ is deduced from the relation $\alpha_{0,exp} = \frac{s}{\pi \Delta \nu}$ assuming pressure broadened lines (Sigrist, 1994). Unlike the near-IR results, the measured maximal absorption $\alpha_{0,exp}$ is in good agreement with the calculated value from the tabulated intensity S and the air-broadening coefficient $\gamma_{air}$ yielding to $\alpha_{0,th} = 1.23 \times 10^{-5}$ cm$^{-1}$. No doubt that in this case, the rotational parameters of N$_2$O are more reliable and here the slight difference between the two values are due to an absorbance averaged on the full line profile compared to an absorption $\alpha_{0,th}$ calculated for the maximum of the rotational line.

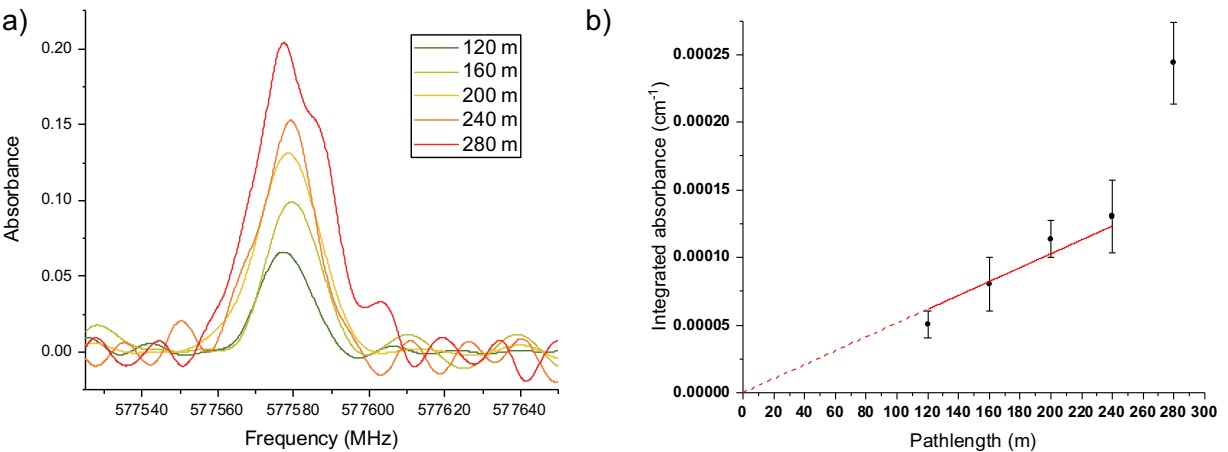

**Figure 5.** (a) Absorbance of the R(22) rotational line of N$_2$O in its ground state measured with the AMC THz source in the 577.58 GHz region for 6 different pathlengths in MULTICHARME ($\chi_{air}$ = 400 ppmv, P = 4 mbar). (b) Linear fit of the of pathlength dependence of the integrated absorbances instrumentally weighted on the error bars deduced from the Lorentzian fits of the absorbances shown in Fig. 5a.

## 3.2 Specificity of the THz measurements

Compared to IR rovibrational spectroscopy, rotational THz spectroscopy presents several advantages and disadvantages for the monitoring of atmospheric compounds in an ASC such as CHARME.

The main advantage deals with the selectivity of the technique at low pressure. Indeed, THz rotational linewidths have two main contributions: the temperature dependent Doppler broadening and the pressure and temperature dependent collisional broadening. The first contribution is the residual source of broadening at low pressure (below 1 mbar), the associated linewidth never exceeds few MHz at THz frequencies giving rotational spectroscopy a much better selectivity than that obtained in IR gas phase rovibrational spectroscopy especially for the light stable and reactive polar atmospheric compounds (De Lucia, 2010). Moreover we have demonstrated the ability of gas phase THz rotational spectroscopy to perform absolute quantification without any calibration step of targeted gaseous pollutants in complex chemical mixtures including both gases and particles (Bigourd et al., 2006, 2007; Mouret et al., 2013). Several measurements performed in realistic gas phase media contaminated with particles demonstrate that THz spectroscopy with submillimeter and mm-wavelengths is less sensitive to scattering than shorter wavelength IR and UV spectroscopy. Finally, due to the capabilities of the RF synthesizer driving the AMC THz source, the acquisition times (typically hundreds of ms) are short and a time-resolved quantification providing kinetic parameters is also possible and simplified by using the THz electronic sources  (Omar et al., 2015).

Despite these different advantages, some difficulties have to be underlined concerning THz monitoring of trace gases in ASCs: (i) first of all, the output power level of the THz sources are significantly smaller than those of optical IR sources : as shown, in Fig.3, the THz measurements are performed at the µW level, at least two orders of magnitude lower than the power available in the near-IR measurements, that affects, inevitably, the sensitivity of the detection scheme; (ii) the price to pay for maintaining an excellent selectivity is to carry out measurements at low pressures representative of the pressure levels of the upper atmosphere and a selective detection of rotational lines at tropospheric pressures is difficult to imagine; (iii) Finally, optical pathlengths between 120 to 240 m produce standing waves with free spectral ranges (FSR) between 625 kHz to 1.25 MHz very close to the linewidths of the measured rotational absorptions. These standing waves strongly affect the baseline and the measured line profiles as observed in Fig. 4a. In Sect. 3.2, we demonstrate some possibilities offered by THz spectroscopic measurements in MULTICHARME taking into account the different advantages and disadvantages previously mentioned.

### 3.2.1 Analysis of isotopic composition

THz rotational spectroscopy is a powerful technique of detection at low pressure due to its great selectivity allowing to discriminate: (i) polar compounds in complex chemical mixture (Bigourd et al., 2006, 2007; Mouret et al., 2013); (ii) isomers and stable conformers amongst VOCs (Roucou et al., 2018, 2020); (iii) isotopomers of small polar atmospheric compounds in natural abundance (Hindle et al., 2019). For this last point, it has been demonstrated that THz rotational spectroscopy is able to determine relative isotopic abundances of small polar compounds with accuracies of few % (Lou et al., 2019).  In order to highlight the selectivity of THz monitoring in MULTICHARME, we present in Fig.6 some measurements of four different isotopomers of pure $N_2O$, in natural abundance. Table 1 summarizes abundances, line frequencies and intensities tabulated in spectroscopic databases (Pickett et al., 1998; Gordon et al., 2022). For the four isotopomers, the differences between the observed and the JPL frequencies never exceed 500 kHz. In Fig. 6a, for each pathlength the absorption of the R(23) rotational

transition of the most abundant $^{14}N^{14}N^{16}O$ isotopomer is saturated. Nevertheless the equivalent transition for the $^{14}N^{15}N^{16}O$ expected only 43 MHz ($1.4 \times 10^{-3}$ cm$^{-1}$) lower in frequency with an intensity around 260 times weaker is clearly observed and

resolved. The other monosubstituted isotopomers $^{15}N^{14}N^{16}O$ and $^{14}N^{14}N^{18}O$ are also observed with the shortest pathlength L = 120 m at pressure close to 1 mbar (Fig. 6b). An isotopic ratio $[^{15}N^{14}N^{16}O]/[^{14}N^{14}N^{18}O]$ of 1.87 is deduced from the intensities of the two absorption lines plotted in Fig. 6b. This value is sufficiently close to the expected value of 1.83 deduced from the natural abundances in Table 1 to suggest the possibility to use THz spectroscopy with MULTICHARME for the analysis of the isotopic composition of atmospheric trace gases and to detect anomalous isotopic signature, a powerful approach to identify

sources and sinks of pollutants and/or greenhouse gases (Röckmann et al., 2001).

   Finally the discrimination of the N₂O isotopomer rotational lines (especially the lines of $^{14}N^{15}N^{16}O$ and $^{14}N^{14}N^{16}O$ on Fig. 6a) highlights the exceptional selectivity of the THz rotational spectroscopy. Indeed , when the measurements are performed at low pressure (typically stratospheric pressures) the linewidths converge to the Doppler limit. For molecules such as N₂O, the Doppler line-widths, proportional to the line frequencies, typically vary from hundreds of kHz in the THz domain

to hundreds of MHz in the IR and to several GHz in the UV visible. No doubt that THz high-resolution monitoring exhibits a significantly better selectivity compared to the IR/UV. In complex chemical mixtures studied in ASC, the THz method allows to observe and resolve individual molecular signatures even for compounds with close molecular structures (isomers, conformers, isotopomers…). Moreover the high-resolution THz method strongly limits the problem of interference substances in the gas monitoring. With photonic detection in the IR/UV spectral domain, it is generally not possible to resolve individual

rovibrational or rovibronic transitions with the typical instrumental resolutions (e.g resolutions of FTIR spectrometers coupled to ASC are limited to few GHz) and some corrections due to interferences with other species have to be taken into account in the trace gas quantification (Harris et al., 2020).

**Table 1.** Natural abundances of the four most abundant isotopomers of N₂O (Gordon et al., 2022). Tabulated line intensity and
frequency of the R(23) rotational transition, respectively from the JPL (Pickett et al., 1998) and the HITRAN (Gordon et al., 2022) database. Difference between tabulated and measured frequencies in Fig. 6.

| Isotopomer | Natural Abundance % | Line intensity cm$^{-1}$/(molecule.cm$^{-2}$) | Line Frequency (MHz) | obs – calc (MHz) |
|---|---|---|---|---|
| $^{14}N^{14}N^{16}O$ | 99.0333 | $2.9 \times 10^{-22}$ | 602666.49 | 0.41 |
| $^{14}N^{15}N^{16}O$ | 0.3641 | $1.11 \times 10^{-24}$ | 602623.41 | 0.06 |
| $^{15}N^{14}N^{16}O$ | 0.3641 | $1.05 \times 10^{-24}$ | 582320.00 | 0.45 |
| $^{14}N^{14}N^{18}O$ | 0.1986 | $5.22 \times 10^{-25}$ | 568975.30 | 0.25 |

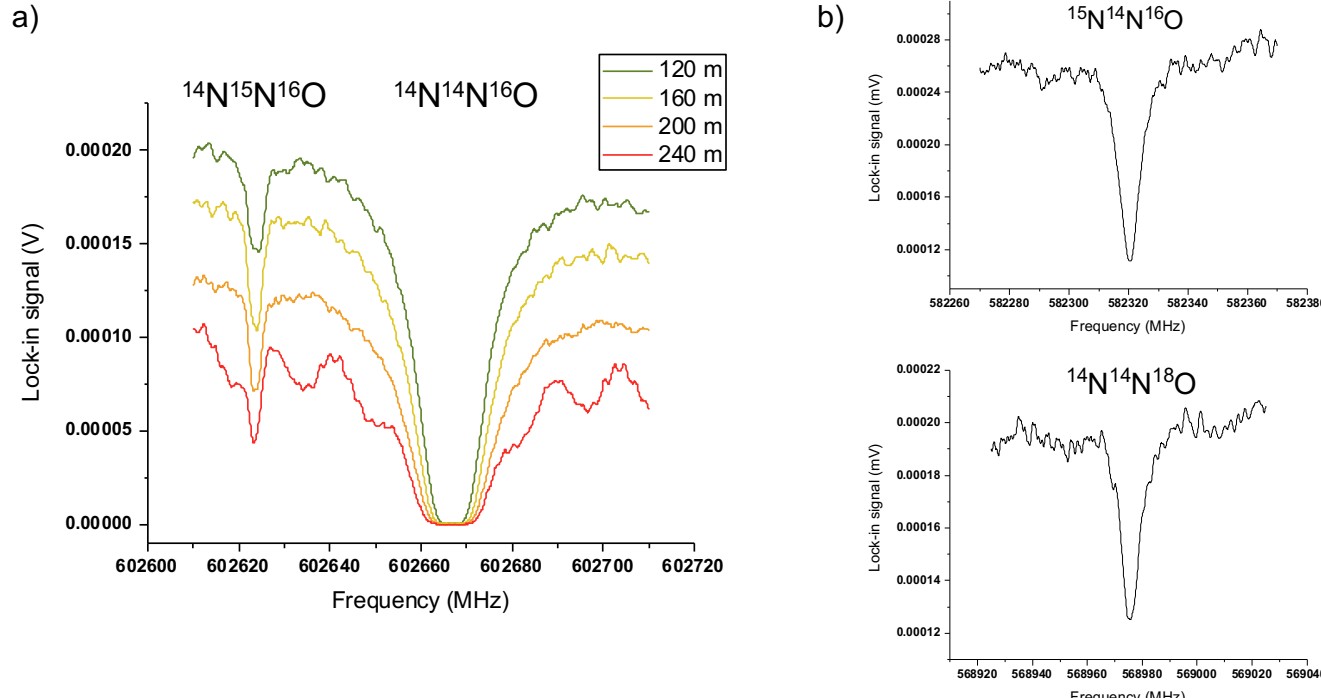

**Figure 6.** THz absorptions of four different $N_2O$ isotopomers in MULTICHARME measured without baseline treatment. (a) R(23) rotational transition measured at P = 0.36 mbar for the most abundant $^{14}N^{14}N^{16}O$ and the less abundant $^{14}N^{15}N^{16}O$ isotopomers; measurements are performed to four optical pathlengths between 120 m to 240 m (in each case the $^{14}N^{14}N^{16}O$ absorption is saturated). (b) Measurements at P ~ 1mbar and L = 120 m, lower in frequency, of the same rotational transition for the $^{15}N^{14}N^{16}O$ and $^{14}N^{14}N^{18}O$ isotopomers.

### 3.2.2 Absolute quantification of stable and reactive atmospheric traces

A priori, all the polar compounds may be detected and quantified from their rotational signatures. In practice, for THz atmospheric monitoring at trace levels, we must opt for the lighter and the more strongly polar compounds with intense and resolved rotational transitions generally listed in the international databases. For these molecules, rotational line frequencies, line widths and line intensities are known with a good degree of accuracy allowing, if the line profile is preserved during the measurement, an absolute quantification without any standard of calibration.

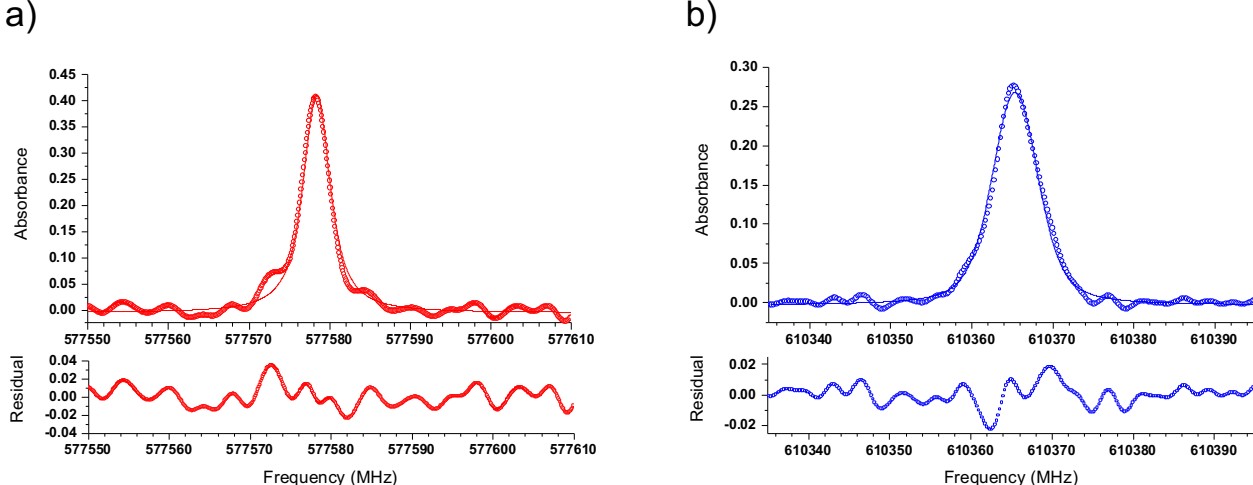

**Figure 7.** THz trace gas quantification of stable $N_2O$ and reactive $O_3$ in MULTICHARME. (a) red circles: R(22) transition of 1000 ppmv $N_2O$ diluted in $N_2$ measured at P = 0.5 mbar (AM: $f_{mod}$=4.5 kHz + FM : $f_{mod}$= 50kHz $\omega_{depth}$=1.44 MHz); red solid line: Fit with a Voigt profile ($\int A(v)dv$=2.4(3) MHz; $v$=577578,36 MHz). (b) blue circles: $25_{1,25} \leftarrow 24_{0,24}$ transition of an unknown quantity of $O_3$ diluted in air produced by an ozone generator measured at P = 1.5 mbar (AM: $f_{mod}$=4.5 kHz + FM : $f_{mod}$= 50 kHz $\omega_{depth}$=1.08 MHz); blue solid line: Fit with a Voigt profile ($\int A(v)dv$=2.3(2) MHz; $v$=610365,35 MHz). The residuals between the Voigt fit and the experimental points are given for both line.

In the present article, we demonstrate this statement in Fig. 7a by fitting with a Voigt profile the absorbance of the R(22) rotational line of $N_2O$ diluted in $N_2$ at 1000 ppmv. Prior to the fit, two baseline treatments have be done in order to reduce the oscillations due to standing waves occurring in MULTICHARME: first of all, due to the capacities of the RF synthesizer, we have applied simultaneously to the amplitude modulation (AM) a rapid frequency modulation (FM) with a depth exceeding the FSR of the interaction length allowing a partial minimization of the effects of the standing waves. Next, during the post treatment of the recorded signal leading to the absorbances shown in Fig. 7, a FFT filter was used *to further reduce the rapid oscillations* of the baseline. This treatment is described in Fig. S5 of the Supplement. We have been careful that the post processing does not affect the line shape and hence the error of the resulting number density. The number density of absorbing $N_2O$ molecules in molec.cm$^{-3}$ is directly deduced from the relation: $N = \frac{\int A(v)dv}{S \times L}$ with for the numerator $\int A(v)dv$ corresponding to the integral, in cm$^{-1}$ unit, of the fitted absorbance by a Voigt profile and for the denominator the product of the line intensity $S$ in cm$^{-1}$/(molec.cm$^{-2}$) tabulated in HITRAN (Gordon et al., 2022) with the pathlength $L$ in cm. Finally the mixing ratio $\chi$ is deduced by: $\chi = \frac{N}{P} k_B T$ with P and T, the pressure and the temperature in CHARME during the measurement. In Fig. 7a, an integrated absorbance of 2.4 ± 0.3 MHz was fitted giving a $N_2O$ number density of N = (1.4 ± 0.2) x 10$^{13}$ molec.cm$^{-3}$ and a mixing ratio 1140 ± 160 ppmv. Taking into account the uncertainty of the fit, mainly due to the remaining

380 baseline oscillations, the density number estimated by the absolute quantification procedure is in agreement with the value of the standard gas used. Based on the previous method, an another example is given in Fig. 7b with the quantification of unknown quantity of ozone in CHARME. Ozone is a key compound in atmospheric chemistry, both in the troposphere and stratosphere (Finlayson-Pitts and Jr, 1999) and a real time in situ monitoring of reactive ozone is very interesting for numerous ozonolysis reactions occurring in our atmosphere especially with VOCs. In our study, ozone was produced at atmospheric pressure by a

385 generator (Air Tree Ozone Technology C-L010-DTI) which converts $O_2$ into $O_3$ from zero air exposed to a high voltage corona discharge. Based on the calibration of the ozone generator, performed with a photometric $O_3$ analyzer, and the injection time (90 min), the ozone volume ratio introduced in CHARME was estimated around 500 ppmv. Then the ASC was pumped down (in 45 min) to 1.5 mbar and the THz spectrometer was used to detect and quantify $O_3$ traces from individual rotational transitions. The $O_3$ wall losses occurring during the ozone introduction as well as during the pumping procedure contribute to

390 reduce the ozone mixing ratio to an unknown lower value which will be measured by THz spectroscopy. Ozone is an asymmetric rotor with a large number of rotational transitions in the THz domain and its rotational frequencies and intensities have been determined with accuracy in the THz domain (Colmont et al., 2005; Birk et al., 1994). The $25_{1,25} \leftarrow 24_{0,24}$ transition centered around 610 GHz with a tabulated intensity of S = 4.035 x $10^{-22}$ cm$^{-1}$/(molec.cm$^{-2}$), the most intense on the source's band emission, was chosen for this reason. A mixing ratio of $258 \pm 22$ ppmv was deduced from the fit of a Voigt profile to the

395 line presented in Fig. 7b. This value is around two times lower than the initial concentration injected in the ASC at atmospheric pressure. This difference is due to the losses on the chamber walls during the ozone injection and the pumping times from atmospheric pressure to 1.5 mbar (135 min.). In Sect. 3.2.3, we show how to characterize at low pressure the kinetics of the $O_3$ losses on the CHARME walls by THz monitoring.

In order to determine the limit of detection (LOD), we have considered the baseline oscillations as our detection noise and the

400 LOD as the concentration obtained with a signal to noise (S/N) equal to 1. Both for $N_2O$ and $O_3$, the S/N of Fig. 7 are estimated to 15 by taking as signal, the maximum amplitude of the rotational line and as noise, the maximum of the amplitude of the residual away from the line. Therefore LOD of around 75 ppmv and 15 ppmv may be estimated respectively for $N_2O$ and $O_3$. These LOD are slightly lower than the mixing ratio errors obtained with the uncertainties on the fitted area. For $N_2O$, the LOD obtained by THz spectroscopy in this study are more than three orders of magnitude larger than the LOD on the strongest mid-

405 IR rovibrational bands by Tunable Diode LAser Spectroscopy (TDLAS) even with measurements at low-pressure (Hoor et al., 1999). With the THz method, for instance the LOD is limited to 15 ppmv. Using the Incoherent BroadBand Cavity-Enhanced Absorption Spectroscopy in the visible domain, a LOD of 120 ppbv was obtained in the Dunkirk ASC at atmospheric pressure (Wu et al., 2014). In order to improve the sensitivity of the THz method, we have to correctly model the baseline and to remove its variations due to multiple interfering stationary waves in MULTICHARME. A work is under progress in this goal.

### 3.2.3 THz monitoring of the ozone decay in CHARME

In Fig. 8, we demonstrate the ability of the THz source coupled to MULTICHARME to monitor at low pressure the ozone reactivity in CHARME. To achieve, the same $O_3$ $25_{1,25} \leftarrow 24_{0,24}$ rotational transition centered at 610365.35 MHz was targeted and measured during 12 hours, over a frequency range of 60 MHz, each 3 minutes. 240 absorbance spectra were obtained and their time evolution as a 3D plot is shown in Fig 8a. For each spectrum, we have repeated the baseline treatment and the line profile fit described in Fig. 7b. Then the data treatment to obtain the 240 absorbances of Fig. 8a and the 240 mixing ratios of Fig. 8b has been batch processed with the Origin Software. As shown in Fig. 8b, the ozone concentration decreases exponentially from 230 ppmv to 15 ppmv in 12 hours. We have considered the first-order kinetics of the ozone decay due mainly to the ozone losses on the chamber walls. The concentration decrease was fitted using the exponential law $[O_3]_t = [O_3]_0 e^{-t/\tau_{O_3}}$. A lifetime $\tau_{O_3}$ of $3.4 \pm 0.1$ h was deduced from a fit weighted on the instrumental errors corresponding here to LODs estimated with the same method as explained in the previous subsection with the Fig. 7b. In the present case, we can see that for ozone concentrations lower than 50 ppm, the lower error bars point to zero or negative values indicating that the LOD is reached as this level of concentration.

The losses of ozone in CHARME have already been investigated at atmospheric pressure with a UV-photometric analyzer (Thermo Scientific 49i; λ = 254 nm). Using several initial concentrations, from 0.7 to 4.8 ppmv, the $O_3$ lifetimes $\tau_{O_3}$ were deduced from the first-order kinetics $O_3$ wall loss reactions and varied from 6.2 to 13.8 h (Fayad, 2019), depending on the cleanliness of the chamber walls which can change for different initial concentrations of ozone. Compared to the previous measurements performed in CHARME with UV photometry analyzer, a shorter lifetime was determined by our low pressure THz measurements. Itoh et al. have developed and experimentally verified a physical model allowing to understand the pressure and the wall material dependencies of the ozone-to-wall loss rate in a cylindrical tube (Itoh et al., 2004, 2011). They showed that the variation of the ozone lifetime with the pressure due to wall losses can be reproduced by the equation (1):

$$\frac{1}{\tau_{O_3}} \sim \frac{D_e}{P} f(a, l, \beta) + kNP \qquad (1)$$

where $P$ is the pressure, $N$ is the molecular density, $f(a, l, \beta)$ is a function depending on geometrical and surface properties of the chamber ($a$ and $l$ are the radius and the length of the cylinder, respectively and $\beta$ is a surface parameter), $D_e$ is an equivalent diffusion coefficient giving the magnitude of the surface loss rate of ozone according to the material and $k$ is a loss rate coefficient due to collisions with oxygen (Itoh et al., 2011). The pressure measurement conditions in THz rotational high-resolution spectroscopy are typical for chamber cleaning activities. Under these conditions it is known that the $O_3$ loss in the chamber is dominated by wall reactions and not by reactions with $O_2$. It corresponds to the first term of Eq. (1) and, therefore, as it is shown by our results, a decrease of the lifetime at low pressure was expected due to the reinforcement of the losses by the ozone diffusion on the walls chamber. That is why these conditions are chosen to get rid of impurities, such as VOCs, on the chamber wall.

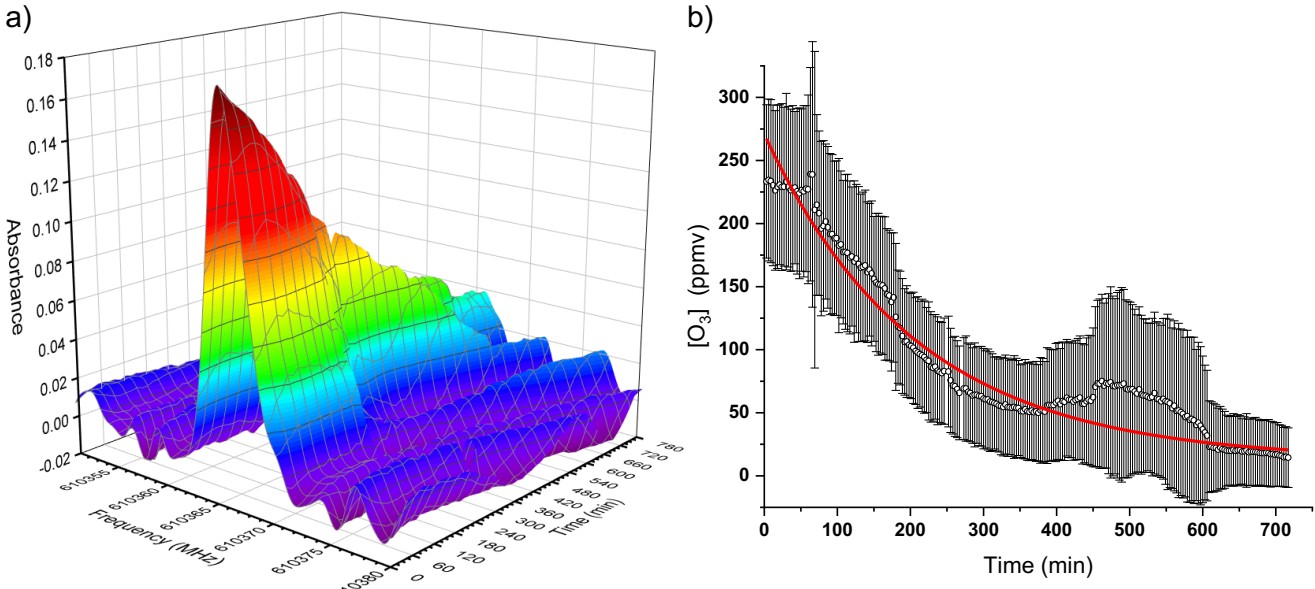

**Figure 8** : THz monitoring of the ozone decay in CHARME measured for 12 h at P= 1.1 mbar. (a) : 3D plot of absorbance spectra measured each 3 min of the $O_3$ $25_{1,25} \leftarrow 24_{0,24}$ rotational transition as a function of frequency (in MHz) and time (in min). (b) : First-order exponential fit of the ozone concentration decay in CHARME deduced by batch processing of the absorbances shown in 8a. A lifetime $\tau_{O_3}$ of $3.4 \pm 0.1$ h is deduced from a weighted fit using the LODs as instrumental errors.

**4. Conclusions**

We have developed for the Dunkirk ASC CHARME a Chernin-type multi-pass cell allowing to perform spectroscopic measurements over approximately three decades of frequencies from the submillimetre-wave spectral domain to the near-IR. In this study, the performances of MULTICHARME have been characterized in the near-IR region using a diode laser source oscillating around 1.4 μm and in the THz region around 600 GHz with a compact and versatile AMC. Benefiting from a base

dimension of 5 m, MULTICHARME allows to reach very long optical paths for absorption spectroscopy from 120 m to 280 m in the THz domain and to 480 m. in the IR. The output powers have been measured for the different pathlengths and an effective mirror reflectivity better than 94% has been measured both in the THz and in the near-IR. By targeting rovibrational and rotational transitions of the $N_2O$ greenhouse gas, the linearity of the integrated absorbances has been checked and experimental values of the maximal absorption coefficient were deduced and compared to the expected values deduced from

the tabulated spectroscopic parameters. The THz monitoring of atmospheric compounds presents some specificities in comparison with other spectroscopic techniques: As example, the measurements of the rotational lines of the most abundant isotopomers of $N_2O$ highlight the exceptional selectivity of the technique which should be able to detect anomalous isotopic fractionation. Moreover the rotational absorbance allows an absolute quantification of the absorbing compound without standard of calibration. The demonstration was performed on stable $N_2O$ and reactive $O_3$ greenhouse gases. According to the

measured S/N ratio, the LOD are limited to 75 ppmv for $N_2O$ and 15 ppmv for $O_3$ due to the baseline oscillations involved by numerous standing waves occurring in MULTICHARME. A work is under progress to characterize how these standing waves affect the THz detected signal in the modified Chernin cell and to study the possibility to correctly model the THz baseline oscillations. This step is required to improve the sensitivity of the method in order to reach subppmv LOD for most of small polar atmospheric molecules showing intense rotational transitions at THz frequencies. We have also to think about the possibility to couple in the future a THz cavity-ringdown spectrometer to CHARME (Hindle et al., 2019), by this way, we hope to be competitive with IR and UV-visible techniques in terms of LOD. Finally, THz monitoring has been used to quantify, at low pressure, the decay of ozone in CHARME. The ozone lifetime of $3.4 \pm 0.1$ h deduced at low pressure in the chamber by THz spectroscopy is shorter than those obtained in previous measurements at atmospheric pressure using a UV photometry analyzer. At low pressure, the ozone losses by diffusion on the ASC are accentuated.

This work demonstrates that THz monitoring is able to quantify gaseous compounds in an ASC such as CHARME, and will allow the determination of kinetic rate coefficients as well as reactional pathways for targeted atmospheric processes. In the future, we plan to couple a Fourier Transform interferometer to MULTICHARME allowing to study the tropospheric reactivity of VOCs at medium resolution ($0.5$ cm$^{-1}$) using vibrational spectroscopy on a broadband spectral range from the far to the near-IR domains ($20$ cm$^{-1}$ - $8000$ cm$^{-1}$). Moreover, THz high resolution rotational spectroscopy will be used in CHARME for the study of the stratospheric processes at low pressures (few mbars) and low temperatures ($T < -20°C$ obtained by cryo-cooling). In particular the chemistry of stable and unstable halogenated species involved in the catalytic destruction of stratospheric ozone are good candidates for these future experiments since the rotational transitions of the stable HX, $CH_3X$ and unstable OX (X=F, Cl, Br, I) compounds lie in the THz domain and are sufficiently intense for their monitoring at trace levels (Pickett et al., 1998).

*Data availability:* Data are available upon request to the corresponding author

*Supplement.* Figures S1, S2, S3 and S4 are available online at:

*Author contributions.* AC was involved with the supervision and conceptualization. JD, EF, PK, WZ, BF and AC contributed to the conception of MULTICHARME. The measurements in CHARME were performed by: JD, EF, JB, NH, FH and AC. JD, JB, RB, GD and AC contributed to the data curation. AC wrote the manuscript with some contributions of JD, FH, NH and CC. The figures were plotted by JD, JB, PK, BF and AC. All the authors contributed to the paper discussion and revision. AC, EF, GM and CC were involved in the funding acquisition. The LPCA towards CHARME is an associated partner of the ATMO-ACCESS European facility.

*Competing interests.* The authors declare that they have no conflict of interest

*Acknowledgements.* We are grateful to the logistics department of the Dunkirk University Management Center for their help in installing the MULTICHARME flanges. Marc Fourmentin is also thanked for his help in the graphical abstract conception.

*Financial Support.* This work was supported by the CaPPA project (Chemical and Physical Properties of the Atmosphere) funded by the French National Research Agency (ANR-11-LABX-0005-01) and the CLIMIBIO program supported by the Hauts-de-France Regional Council, the French Ministry of Higher Education and Research and the European Regional Development Fund. J.D and J.B were funded respectively by CLIMIBIO and the Pôle Métropolitain de la Côte d'Opale. MULTICHARME was founded by the Research Quality Bonus of the University of Littoral Côte d'Opale, the Optics, Photonics, Applications Lasers (OPAL) network and the IRenE program.

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
