# Peer review of "MULTICHARME: A modified Chernin-type multi-pass cell designed for IR and THz long-path absorption measurements in the CHARME atmospheric simulation chamber."

_Atmospheric Measurement Techniques, 2021_

## Referee Comment (RC1)

**Review of Preprint:**

**Article**: Decker, J., Fertein, É., Bruckhuisen, J., Houzel, N., Kulinski, P., Fang, B., Zhao, W., Hindle, F., Dhont, G., Bocquet, R., Mouret, G., Coeur, C., and Cuisset, A.: MULTICHARME: A modified Chernin-type multi-pass cell designed for IR and THz long-path absorption measurements in the CHARME atmospheric simulation chamber, Atmos. Meas. Tech. Discuss. (2021), https://doi.org/10.5194/amt-2021-399.

The authors describe first measurements and the spectroscopic characterization of a modified Chernin-type multi-pass cell, which has been designed for wavelengths in the IR and THz region. The multipass cell has been set up as part of the detection equipment of the CHamber for Atmospheric Reactivity and Metrology of the Environment (CHARME) in Dunkirk. The combination of this type of multipass cell and a low-pressure atmospheric chamber is novel, and the data presented inspire confidence in the method.

The cell enables path lengths between 120 and 480 m (540 m) in the IR and between 120 m and 280 m in the THz region. Proof-of-principle measurements have been performed with $N_2O$ and $O_3$ whose mixing ratios were established. While the sensitivity of the method for those species is modest, its selectivity is a strength, enabling the potential to study less common (polar) VOCs in the future. While the authors address detection limits, applicability of the detection approach, its selectivity and suitability for kinetic studies based on the time resolution during an $O_3$ reactivity study, I am missing a general comparison with other methods and a classification of the method among other (spectroscopic) approaches in different wavelength regions.

The authors discuss some advantages and drawbacks of their approach. They also describe experimental difficulties and how they were overcome, however, more attention to detail, e.g. in the establishment of the LOD or in the discussion of systematic errors, would be helpful.

The overall presentation of the work is well structured and clear, however, in several places some confusion may arise due to the wording used. The authors give sufficient credit to related work and with a few exceptions most references appear appropriate.

I recommend acceptance of this manuscript for publication after addressing the observations in this review; I consider the large majority of comments as minor.

**Title**
L3. Delete the "full stop" at the end of the title.

**Abstract**
L20. Improve the sentence: "Moreover, a THz monitoring at low pressure of the ozone decay in the chamber has been performed."
L25. "to reach atmospheric trace levels."
L25/26. Improve the sentence.

**Introduction**
L30. large panel -> large variety
L31. apparatus -> apparati
L31. laboratory developed -> custom-designed
L36. quantification yielding to kinetic -> quantification of kinetic
L40. Use a uniform way to denote pathlength throughout the manuscript: either "pathlength" or "path length"
L43. define VOC
L45. The classic reference after "White-cell" would be good
L50. Valence -> Valencia

L55. Even though specific, I would give also credit the older publication here:
S. M. Chernin and E. G. Barskaya, Optical multipass matrix systems, Appl. Opt. 30, 51-58 (1991).
L58. Delete "than" before "100 ppb". Parts per billion should be stated as "by volume", i.e. ppbv
L59. Full stop is in the wrong place.
L62. …to control the propagation of more divergent beams over long distances…
L63. Far-IR -> far-IR;      trace gases -> trace gas
L66. comma in reference
L67. weak -> small
L69. comma in reference
L73. If the results were preliminary, they should not be published here. Delete "preliminary".

**Experimental setup and Methodology**
L81. Comma after briefly.
L82. made in -> made of
L85. Depending on power and geometry of the fan system and depending on the nature of the reactive species being studies, the stirring of the gas mixture can lead to an increase of wall losses of the reactive species and not to a homogenization. I think this statement may require a reference concerning a study of the effect of the fans or should be phrased more carefully.
L89. …filled with purified and dried air at the required pressure using …
L91. The Baratrons only measure the pressure but do not control it. Is the MKS PR400B connected to a pressure controlling flow meter? If that is the case then that should be pointed out, otherwise there is no component here that actively "controls" the pressure as claimed.
L94. "…accommodate the MULTICHARME optical …"
L96. *et al.*  should be not italics. I would cite the original paper here also:
S.E. Fiedler, A. Hese, A.A. Ruth, "Incoherent broad-band cavity-enhanced absorption spectroscopy", Chem. Phys. Lett. 371, 284-294 (2003).
L104. so as to anticipate -> to account for
L106. substrate was -> substrates were
L110. A supporting reference from a previous Anhui publication would be good here.
L112. DN 450 access ports (A1 and A2 in Figure S1)…  insert a cross-reference to the supplementary material.
L115. located on opposite ends of the cylindrical chamber.
Caption Figure 1.  …two mirror's blocks…   -> …two mirror assemblies…
objective's block -> objective block
field's block  ->  field block
L118. What kind of "static analysis" was performed? Finite element calculation? What conditions (force field) were assumed? Somewhat more detail is required here or an appropriate reference should be stated.
L127. moving  ->  movement
L127/128. Improve sentence: "The five mirrors' configuration is easy to align, with very good stability to vibrations, and gave variable rows with even column images on the field mirrors."
L130. A pathlength of 540 m is claimed here, however, measurements are only shown up to 480 m.
L132. Extended  -> External
L133. Delete "power" after 80 mW
L134. Name the photodiode and give some specs. Ge, InGaAs? Bandwidth?
Name the oscilloscope and give some specs (e.g. vertical resolution, sample rate, max frequency)
L135. DL source  ->  laser     or      -> ECDL
L142-144. Improve this sentence. To the accuracy of what parameter does the value of $10^{-7}$ refer to? The wavelength range was calibrated using the Burleigh wavemeter with a specific accuracy? Can you give an absolute value?
Caption Figure 2: performed -> established; Extended -> External; HRFZ needs to be explained
L159. Fix the way the reference is cited.
L162/163. A and $R_{eff}$ should be in italics.
L164. How were the error bars determined? Insert a cross-reference to the caption of Figure 3.

Figure 3. I would plot a power law always in a double logarithmic graph rather than using linear axes. The right axes are missing in panels (a) and (b).

Caption of Figure 3. $R^n$ -> $R_{eff}^n$ ;   "… , with $\sigma_i$ being the error bar…"; "… of the controller sensor…".  Can the THz power fluctuations also be quantified?

L175. "…using high resolution …"   What are realistic media? What is meant by that?

L179. Can more information be given on the code from LightMachinery Inc.?

L185. Detectors

L189. "…as a function of frequency."

L191. "…He-Ne laser wavelength…"

L198. Delete "important"

L199-201. Rephrase – these sentences are rather casual and should be more factual.

L201. Fix the referencing.

L204. "…exceeded"

**Results and discussion**

L214. molecule test  ->  test molecule

L217. their -> the    or   its

L218 & 221. Fix referencing (italics)

Figure 4 and L236. The unit on the ordinate of Fig. 4(b) and in Line 236 seems incorrect as far as the axis title is concerned. Depending on what variable the absorption coefficient is integrated over (frequency or wavenumber), the unit should not remain [$cm^{-1}$], which is the unit of the absorption coefficient itself. What is probably meant here is the "integrated absorbance", then the unit of [$cm^{-1}$] is correct, if integrated over wavenumber.

Caption of Figure 4 & L237 &L238. gaussian -> Gaussian

L236. Is the linear regression going through the origin or was it forced through zero? This is difficult to see in Figure 4(b). With a non-zero intercept the slope may change somewhat. Remedy: State the fit function explicitly.

L238. Unit of slope okay if integration over absorbance.

L240. Kwabia Tchana et al., 2013

L241. "…estimated to be …"

Based on Figure 4(a) the estimated HWHM seems to be somewhat too small. FWHM seems to be more like ~0.028 $cm^{-1}$.

L241. The equation in that line requires more explanation. How was it derived?

I find alpha_0,exp = s*sqrt(ln2)/(delta_nu*sqrt(pi)), if what was called "integrated absorption coefficient" is indeed "integrated absorbance". If a HWHM of 0.014 $cm^{-1}$ is used this results in a similar value as stated, i.e. $3.92 \times 10^{-6}$ $cm^{-1}$. If the original HWHM is used one finds a value that is even larger, i.e. $5.50 \times 10^{-6}$ $cm^{-1}$.

What would be of interest here also is to compare this value with the measured alpha_0,exp, averaged for all 8 different pathlength measurements.

L254. Where does the integrated line intensity come from? How was this estimated? There is a reference needed here. It also says "experimentally measured". By whom? In this work?

L257. "…to the Doppler one…"  -> "… to Doppler broadening…"

L260. I am again a bit confused here as before. How can an integrated absorption coefficient be deduced by integrating over the absorbance (in each case integrated over frequency or wavenumber)?

L265. "…estimated to be …"

L265. How was alpha_0,exp calculated here? A Voigt profile is used for the description of the absorbance of the measured line. What assumption was made?

L268. Give a reference for the line intensity S.

L280: delete "these last years"

L284: "to scattering". Replace "radiations" by "spectroscopy". Change the word "agility"

L285: "rapid"  -> "short"

L288: "gas traces"  -> "trace gases"

L289: "compared to the optical IR one"  ->   "are significantly smaller than those of optical IR sources:"

L306: "for four"  -> "of four"

L310: $1.4 \times 10^{-3}$ cm$^{-1}$

Figure 6b: Labels on axes are very small

L322: "to" -> "and"

Table 1: Use proper scientific notation in column 3.

L337: "…by fitting.." what? A 'Voigt profile'? The function that is fitted to the data should be stated here ("…by fitting a Voigt profile…"). Moreover, in Figure 7 the absorbance spectrum of the R(22) of $N_2O$ line is shown. What do the authors mean by "integrated intensity" in Line 337?

L339: Replace the word "agility" with something meaningful.

L338-342: It should be argued or shown that the "two baseline treatments" have no effect on the line shape and width. A comparison of results with and without the treatment could be shown here, since data manipulations like FFT filtering affect the line shape and hence the error of the resulting number density. A systematic error discussion could be included here.

L343: concentration "N" -> number density

Figure 7: Red and blue circles (or panels) seem to have been mixed up. It would be meaningful to show the fit residuals in panel below the main figures. The unit of the integrated absorbance is stated in MHz, however as per the main text (L344) this should be wavenumbers. Please be uniform in your notation.

L354: What is meant by "… an integrated absorption of 2.4 MHz was fitted …"? Concentration -> number density.

L354: use scientific notation for the value of the number density.

L356: rephrase "..provides the level of dilution stipulated in the calibrated gas."

L357: delete "an"

L361: A -> An

L362/363: Rephrase the sentence: According …, This… analyzer.

L363: "pumped up" -> "pumped down"

L367: Use scientific notation in for the value of the line strength S.

L368: "…from the fit of the line with a Voigt profile …" -> "…from the fit of a Voigt profile to the line …"

L375. The pressure should be explicitly state here and not only in the caption of Figure 8. State the chamber conditions better.

L376: "into" -> "in". " In this aim" ? 610365.35 (no comma, like in French)

L377: "during" -> "for" Separate the text in L376 – 378 into 2 sentences. Rephrase.

L378: "spectra" -> "spectrum"; "reproduced" is not the right word here -> "repeated".

L380. "Concentrations" -> "mixing ratios"

L381. "decrease" -> "decreases"

L382. "due mainly due" ? ; "losses" -> "ozone losses" ; "walls chamber" -> "chamber walls"

L382-385. Split the sentence "The concentration decrease was fitted…" into two or three sentences. What is meant by "…a fit weighted on the estimation of the limit of detection"? This is not clear. The LOD was estimated based on a signal to noise ratio of 1; the authors should say more here. Explain better how the maximal amplitude of the baseline oscillations was determined. Over what spectral region, for what time in the measurement series. What was the maximal signal, S? The integrated absorbance or the max value of the absorbance. In the caption of Fig 8. The authors refer to the "absorbance area". This is not clear to me.

L386. The LOD should be properly stated; i.e. $50 \pm$ ?? ppmv. …"we are very close to this limit" is too casual. What is the acquisition time for this LOD, is it 3 min?

Figure 8 caption: Caption should be non-centered. "Fig. 8:" -> "**Figure 8.**" "during" -> "for". The panels (a) and (b) should not be labelled "Fig.8a" and "Fig. 8b". Rephrase the sentence "3D plot gathering …" use different wording. "Weighted on the LODs"?

L395. walls -> wall

L396. "cleaning state"? Cleanliness?

L399. "ozone walls" -> "ozone-to-wall"

L400. "show" -> "showed"

Since the concentrations in the current experiment are significantly higher it is not clear how meaningful this comparison to the work by Itoh *et al.* is. Itoh *et al.* measured from a pressure of 6.7 mbar, which is not even as low as in the present study it seems? The conditions in the present paper

are typical for chamber cleaning activities. Under these conditions it is known that the $O_3$ loss in the chamber is dominated by wall reactions and not by reactions with $O_2$. That is why these conditions are chosen to get rid of impurities, such as volatile organic compounds, on the chamber wall.
L405. "))" typo
L406/407. Ozone being generated at atmospheric pressure? I thought the chamber was kept at low pressure in these experiments (see L375.).
L407. "loses" -> "losses"

**Conclusions**
L415. "…in the THz region"
L417. Measurement at 540 m not shown - no experimental evidence in this paper. The authors may want to include a measurement at 540 rather than just showing an additional long path pattern in the supplementary material.
L426. More consideration should be given to the detection limit(s) in this article; "a few tens of ppmv" is too unspecific.
L433. "such as CHARME"
L436. "middle" -> "medium"

L450. "contributed"
L458. "for its help" -> "for his help"
L463. "JD and JB …"

**References**
Many DOIs are missing.
Make the references in the list more uniform.
In the text the citing of references should be uniform. Sometimes et al. is italics, sometimes not, sometimes with comma, sometimes without.

**Supplement**
The aspect ratio of the photographs in Figure S1 seem non uniform.

---

## Author Comment (AC1)

**Response to RC1:**

*The authors describe first measurements and the spectroscopic characterization of a modified Chernin- type multi-pass cell, which has been designed for wavelengths in the IR and THz region. The multipass cell has been set up as part of the detection equipment of the CHamber for Atmospheric Reactivity and Metrology of the Environment (CHARME) in Dunkirk. The combination of this type of multipass cell and a low-pressure atmospheric chamber is novel, and the data presented inspire confidence in the method.*

First of all, we would like to thank RC1 for their thorough and meticulous reading of the manuscript which has allowed us to significantly improve the quality of our article. We really appreciate the trust he has placed in our measurements and interpretation. In addition, it is very important for us that he noticed the novelty of our approach with the first high-resolution THz measurements at low pressure in an Atmospheric Simulation Chamber (ASC) such CHARME.

*The cell enables path lengths between 120 and 480 m (540 m) in the IR and between 120 m and 280 m in the THz region. Proof-of-principle measurements have been performed with $N_2O$ and $O_3$ whose mixing ratios were established. While the sensitivity of the method for those species is modest, its selectivity is a strength, enabling the potential to study less common (polar) VOCs in the future. While the authors address detection limits, applicability of the detection approach, its selectivity and suitability for kinetic studies based on the time resolution during an $O_3$ reactivity study, I am missing a general comparison with other methods and a classification of the method among other (spectroscopic) approaches in different wavelength regions.*

We agree with the reviewer: the main weakness of the THz method is, for the moment, a modest sensitivity for trace gas monitoring compared to several IR and UV photonic techniques but this weakness is counterbalanced by its selectivity at low-pressure when the rotational linewidths are Doppler limited. Always in agreement with the reviewer, the second advantage of the THz monitoring is the capability to detect *a priori* all the polar atmospheric VOCs and to unambiguously discriminate them with their rotational signatures which would be difficult with vibrational IR and electronic UV bands measured at lower resolution. Finally, due to the capabilities of the RF synthesizer driving the AMC THz source, the acquisition times (typically hundreds of ms) are short and a time-resolved quantification providing kinetic parameters is also possible and simplified by using the THz electronic sources (Omar et al., 2015).

RC1 regrets the absence of a general comparison with other methods and a classification of the method among other (spectroscopy) approaches in different wavelength regions. It is true that there is no specific paragraph dedicated to this type of comparison but, nevertheless, the advantages and disadvantages of THz monitoring compared to other photonic techniques are discussed at different places in the discussion and in the conclusion:

- In the last paragraph of section 3.1, the THz selectivity is compared with the IR one: "*the THz rotational linewidth is estimated to be $3.4 \times 10^{-4}$ cm$^{-1}$ (HWHM), around 300 times smaller than the IR rovibrational linewidth highlighting the excellent selectivity of the THz spectroscopy compared to the IR one due to a weaker Doppler broadening.*"

- The first part of section 3.2 is a general comparison of the advantages and disadvantages of the THz rotational spectroscopy compared to IR rovibrational spectroscopy:

*"Compared to IR rovibrational spectroscopy, rotational THz spectroscopy presents …….. and a time-resolved quantification providing kinetic parameters is also possible and simplified by using the THz electronic sources (Omar et al., 2015).*

*Despite these different advantages, ….. These standing waves strongly affect the baseline and the measured line profiles as observed in Fig. 4a"*

- Finally in the conclusions section, the specificities of the THz monitoring compared to other techniques are again mentioned:

*"The THz monitoring of atmospheric compounds presents some specificities in comparison with other spectroscopic techniques: As example, the measurements of the rotational lines of the most abundant isotopomers of $N_2O$ highlight the exceptional selectivity of the technique which should be able to detect anomalous isotopic fractionation. Moreover the rotational absorbance allows an absolute quantification of the absorbing compound without standard of calibration. The demonstration was performed on stable $N_2O$ and reactive $O_3$ greenhouse gases. According to the measured S/N ratio, the reached detection thresholds are limited to few tens of ppmv due to the baseline oscillations involved by numerous standing waves occurring in MULTICHARME."*

In order to take into account the remarks of RC1, additional elements have been added in the manuscript:

-At the end of the subsection 3.2.1:

*"Finally the discrimination of the $N_2O$ isotopomer rotational lines (especially the lines of $^{14}N^{15}N^{16}O$ and $^{14}N^{14}N^{16}O$ on Fig. 6a) highlights the exceptional selectivity of the THz rotational spectroscopy. Indeed , when the measurements are performed at low pressure (typically stratospheric pressures) the linewidths converge to the Doppler limit. For molecules such as $N_2O$, the Doppler line-widths, proportional to the line frequencies, typically vary from hundreds of kHz in the THz domain to hundreds of MHz in the IR and to several GHz in the UV visible. No doubt that THz high-resolution monitoring exhibits a significantly better selectivity compared to the IR/UV. In complex chemical mixtures studied in ASC, the THz method allows to observe and resolve individual molecular signatures even for compounds with close molecular structures (isomers, conformers, isotopomers…). Moreover the high-resolution THz method strongly limits the problem of interference substances in the gas monitoring. With photonic detection in the IR/UV spectral domain, it is generally not possible to resolve individual rovibrational or rovibronic transitions with the typical instrumental resolutions (e.g resolutions of FTIR spectrometers coupled to ASC are limited to few GHz) and some corrections due to interferences with other species have to be taken into account in the trace gas quantification (Harris et al., 2020)."*

-At the end of the subsection 3.2.2:

*"In order to determine the Limit of detection (LOD), we have considered the baseline oscillations as our detection noise and the LOD as the concentration obtained with a signal to noise (S/N) equal to 1. Both for $N_2O$ and $O_3$, the S/N of Fig. 7 are estimated to 15 by taking as signal, the maximum amplitude of the rotational line and as noise, the maximum of the amplitude of the residual away from the line. Therefore LOD of around 75 ppmv and 15 ppmv may be estimated respectively for N2O and O3. These LOD are slightly lower than the mixing ratio errors obtained with the uncertainties on the fitted area. For N2O, the LOD obtained by THz spectroscopy in this study are more than three orders of magnitude larger than the LOD on the strongest mid-IR rovibrational bands by Tunable Diode LAser Spectroscopy (TDLAS) even with measurements at low-pressure (Hoor et al., 1999). With the THz method, for instance the LOD is limited to 15 ppmv. Using the Incoherent BroadBand Cavity-Enhanced Absorption Spectroscopy in the visible domain, a LOD of 120 ppbv was obtained in the Dunkirk ASC at atmospheric pressure (Wu et al., 2014). In order to improve the sensitivity of the THz method, we have to correctly model the baseline and to remove its variations due to multiple interfering stationary waves in MULTICHARME. A work is under progress in this goal."*

-In the conclusion:

*"This step is required to improve the sensitivity of the method in order to reach subppmv LOD for most of small polar atmospheric molecules showing intense rotational transitions at THz frequencies. We have also to think about the possibility to couple in the future a THz cavity-ringdown spectrometer to CHARME (Hindle et al., 2019), by this way, we hope to be competitive with IR and UV-visible techniques in terms of LOD."*

**The authors discuss some advantages and drawbacks of their approach. They also describe experimental difficulties and how they were overcome, however, more attention to detail, e.g. in the establishment of the LOD or in the discussion of systematic errors, would be helpful.**

The sensitivity of MULTICHARME is limited by the complex and strong baseline variations observed at THz frequencies caused by multiple standing waves. To evaluate the LOD we quantify the amplitude of the recorded molecular signal and the amplitude of the baseline variations "noise", both after normalization (sample spectra)/(ref spectra). The baseline variation is determined either by comparing two spectra with an empty measurement cell, or examining a portion of a spectra with no molecular transition. The LOD is then determined by an experimental data close to that limit, weak trace absorption whose concentration is extrapolated by the S/N to give the LOD. The LOD then corresponds to the concentration that would give S/N = 1.

As it was mentioned previously, in order to determine the LOD, we have considered these baseline oscillations as our detection noise and therefore our LOD is the concentration obtained for a S/N = 1. The effect of the baseline oscillations is now clearly visible with the residuals provided in Fig. 7 as suggested by Reviewer 1 and the description of the fitting process in the new Fig. S5 of the Supplement. As example, with the two THz lines shown in Fig. 7 exhibiting a S/N around 15 if we take as signal, the maximum amplitude of the rotational line and as noise, the maximum of the amplitude of the residual away from the line. So, we can estimate the LODs of around 75 ppmv and 15 ppmv respectively for $N_2$ and $O_3$. In the article, in Fig. 7, we have also determined systematically the errors on the fitted area and consequently the errors on the associated mixed ratio (ex: Fig. 7a, $\pm$ 160 ppmv; Fig. 7b, $\pm$ 22 ppmv). These errors are slightly superior to the LODs. We hope that the additional explanations in subsection 3.2.2 allow the reader to better understand our approach concerning the LOD and the systematic errors determination. The sensitivity improvement requires right now to correctly model the baseline variation due to standing waves, this work is under progress and will be the subject of a future publication.

**The overall presentation of the work is well structured and clear, however, in several places some confusion may arise due to the wording used. The authors give sufficient credit to related work and with a few exceptions most references appear appropriate.**

One more time, we are grateful to RC1 for its careful reading. We have taken into account all its suggested corrections concerning several bad wordings. As asked by the reviewer 1, we have improve several sentences:

- L20. The sentence: "*Moreover, a THz monitoring at low pressure of the ozone decay in the chamber has been performed*" is replaced by "*At low pressure the ozone concentration was continuously monitored and its decay characterized.* »
- L25.26 The sentence: " *However, the sensitivity of the THz monitoring needs to be improved to reach atmospheric trace levels. For this purpose, it is necessary to fully understand the origin of the observed baseline variations caused by the complex multiple standing waves present in MULTICHARME.*" is replaced by "*Right now, the accessible detection levels for both compounds are not sufficient to detect both compounds at atmospheric concentrations. A correct baseline*

*modelisation in order to remove its variations due to multiple interfering stationary waves in the Chernin cell is required."*

- **L142-144. *Improve this sentence. To the accuracy of what parameter does the value of $10^{-7}$ refer to? The wavelength range was calibrated using the Burleigh wavemeter with a specific accuracy? Can you give an absolute value?***

$10^{-7}$ refers to the relative accuracy on the wavelength measurement $\frac{\Delta\lambda}{\lambda}$. Yes the wavelength range was calibrated using the Burleigh wavelength-meter with its specific accuracy. The sentence has been replaced by:

*Once the laser was adjusted for the desired operating range, a wavelength calibration was performed (Burleigh WA-1500) with an accuracy better than 4 x $10^{-3}$ cm$^{-1}$.*

- **L199-201. *Rephrase – these sentences are rather casual and should be more factual.***

We propose a new sentence: *"Reaching a pathlength of 280 m with an amplified frequency multiplication chain which is highly divergent source is a significant improvement compared commonly used setups. Extending the pathlength further should be possible for higher frequencies or with more powerful THz sources."*

- **L382-385. *Split the sentence "The concentration decrease was fitted..." into two or three sentences. What is meant by "...a fit weighted on the estimation of the limit of detection"? This is not clear. The LOD was estimated based on a signal to noise ratio of 1; the authors should say more here. Explain better how the maximal amplitude of the baseline oscillations was determined. Over what spectral region, for what time in the measurement series. What was the maximal signal, S? The integrated absorbance or the max value of the absorbance. In the caption of Fig 8. The authors refer to the "absorbance area". This is not clear to me. L386. The LOD should be properly stated; i.e. 50 ±?? ppmv. ..."we are very close to this limit" is too casual. What is the acquisition time for this LOD, is it 3 min?***

We use a weighted fit when the assumption of a constant variance of the errors is not respected. The weighting allows to improve the fit. The idea is simple, we give less weight to the less precise measurements and more weight to more precise measurements when estimating the unknown parameters in the model. With the Origin software, in the Levenberg Marquardt fitting process, we have chosen to perform an exponential fit weighted on the LODs considered as instrumental errors i.e. a weight equal to the LOD^(-2) (see https://www.originlab.com/doc/Origin-Help/FIt-with-Err-Weight).
In subsection 3.2.3 dedicated to the THz monitoring of the ozone decay in CHARME, the error bars correspond also to the LODs estimated for all the measurements (each 3 min.) with the same method explained previously. We think that these error bars could be overestimated due to the batch processing.

According to this remark, the sentence has been rewritten:

*"The concentration decrease was fitted using the exponential law $[O_3]_t = [O_3]_0 e^{-t/\tau_{O_3}}$ . A lifetime $\tau_{O_3}$ of 3.4 ± 0.1 h was deduced from a fit weighted on the instrumental errors corresponding here to LODs estimated with the same method as explained in the previous subsection with the Fig. 7b. In the present case, we can see that for ozone concentrations lower than 50 ppm, the lower error bars point to zero or negative values indicating that the LOD is reached as this level of concentration."*

***L85. Depending on power and geometry of the fan system and depending on the nature of the reactive species being studies, the stirring of the gas mixture can lead to an increase of wall losses of the reactive species and not to a homogenization. I think this statement may require a reference concerning a study of the effect of the fans or should be phrased more carefully.***

The comment of RC1 (Line 85) is related to fan rotation speed, but in our experiments the fans were not activated. Indeed, the fans can only be used at atmospheric pressure (it is now mentioned in the section 2.1). However, we agree that the stirring of the gas mixture can lead to an increase of wall losses of the reactive species and not to a homogenization. This was observed in CHARME for secondary organic aerosols (see Fayad, 2019).

***L118. What kind of "static analysis" was performed? Finite element calculation? What conditions (force field) were assumed? Somewhat more detail is required here or an appropriate reference should be stated.***

The total deformation of the Chernin cell and the corresponding mounting flanges was analyzed under static conditions. The forces that considered were their own gravity and the atmospheric pressure on the outer surface of the flanges when the chamber was in a vacuum state. This last sentence has been added in the revised text.

***L130. A pathlength of 540 m is claimed here, however, measurements are only shown up to 480 m.***

As shown in Fig. 3a, during the IR measurements, we succeed to reach a pathlength of 540 m. corresponding to a matrix configuration of 9 rows and 6 columns (108 reflections) but the IR output power (around 50 μW) was not sufficient to detect the IR rovibrational line. In the manuscript, the longest IR pathlength is limited to 480 m. excepted for the power measurements (Fig. 3a). In the conclusion we also give an upper limit of 480 m. according to a 2$^{nd}$ remark of RC1.

***L134. Name the photodiode and give some specs. Ge, InGaAs? Bandwidth?***
***Name the oscilloscope and give some specs (e.g. vertical resolution, sample rate, max frequency)***

We did it:

*"An InGaAs detector (Thorlabs PDA400) with a typical bandwidth of 10MHz was used for the detection. Spectra were obtained by applying a voltage ramp to the piezo actuator allowing an excursion of 0.17 cm$^{-1}$ around the line center of interest with a repetition rate of 1.3 Hz. The received photodiode signal was averaged by a digital oscilloscope (DSO-X 2002A Agilent Technologies, maximum frequency of 70MHz), the signal was typically accumulated over 16 ramp cycles with a sampling of 12.5 kHz (10 bits of vertical resolution)."*

***L164. How were the error bars determined?***

Here it is simply the error given by the fit. It is now mentioned in the text.

***Figure 3. I would plot a power law always in a double logarithmic graph rather than using linear axes. The right axes are missing in panels (a) and (b). Can the THz power fluctuations also be quantified?***

The figure 3 has been changed in order to provide both linear scale with the power law fit (in blue) and log scale with the linear fit (in red). Of course, both fits lead to the same reflectivity values. The THz error bars correspond to the measured power fluctuations.

***L179. Can more information be given on the code from LightMachinery Inc.?***

The web link is now given in the references.

**Figure 4 and L236. The unit on the ordinate of Fig. 4(b) and in Line 236 seems incorrect as far as the axis title is concerned. Depending on what variable the absorption coefficient is integrated over (frequency or wavenumber), the unit should not remain [$cm^{-1}$], which is the unit of the absorption coefficient itself. What is probably meant here is the "integrated absorbance", then the unit of [$cm^{-1}$] is correct, if integrated over wavenumber.**

RC1 is right, we have used "the integrated absorbance" in $cm^{-1}$ unit and not the "the integrated absorption coefficient" which will be in $cm^{-2}$ unit. This mistake has been corrected all along the manuscript including the Y-axis labelling of Fig.4b and 5b. It answers also to reviewer for its remarks L.260.

**L236. Is the linear regression going through the origin or was it forced through zero? This is difficult to see in Figure 4(b). With a non-zero intercept the slope may change somewhat. Remedy: State the fit function explicitly.**

We have used a linear function and not an affine function for the fits in Fig. 4b and 5b. So yes the regression was forced trough zero. It is now mentioned in the caption and the origins are now visible on Fig. 4b and 5.b.

**Based on Figure 4(a) the estimated HWHM seems to be somewhat too small. FWHM seems to be more like ~0.028 $cm^{-1}$.**

Here, we don't agree with RC1. As it was mentioned in the caption of the Fig. 4, the shoulder observed at low frequency is assigned to the ECDL source which is not perfectly monomode in this region. This shoulder does not belong to the probed rovibrational line and must not be taken into account in the rovibrational linewidth. This explains why we have this smaller linewidth compared to the value proposed by RC1.

**L241. The equation in that line requires more explanation. How was it derived? I find alpha_0,exp = s*sqrt(ln2)/(delta_nu*sqrt(pi)), if what was called "integrated absorption coefficient" is indeed "integrated absorbance". If a HWHM of 0.014 $cm^{-1}$ is used this results in a similar value as stated, i.e. 3.92×10$^{-6}$ $cm^{-1}$. If the original HWHM is used one finds a value that is even larger, i.e. 5.50×10$^{-6}$ $cm^{-1}$.**

The book of M. Sigrist (Sigrist, 1994) was used as reference for the alpha_0, exp equations. This reference is now added in the manuscript. alpha_0 is in $cm^{-1}$ unit, it corresponds well to an absorption coefficient as it was mentioned previously by Reviewer1. We thank the reviewer1 for its remark and we have corrected the *alpha_0,exp* expression for the near-IR measurements where the Doppler dominates the collisional broadening. So, in this case, we agree with reviewer1, *alpha_0,exp = s*sqrt(ln2)/(delta_nu*sqrt(pi))= 3.92×10$^{-6}$ $cm^{-1}$* with delta_nu = *0.014 $cm^{-1}$* (HWHM).

**L265. How was alpha_0,exp calculated here? A Voigt profile is used for the description of the absorbance of the measured line. What assumption was made?**

Compared to the Fig.4 where the near-IR where mainly Doppler broadened, the THz rotational lines in Fig.5 have a very small Doppler line-width (0.54 MHz HWHM). Therefore a lorentzian profile is a good approximation and for this reason the absorption coefficient should be calculated with the relation *alpha_0,exp = s/(delta_nu*pi)*. In the revised version of the manuscript we have updated Fig. 5b using now Lorentzian profiles to fit the absorbances shown in 5a. A linear fit without intercept is obtained from 120 m to 240 m. Compared to the previous version the slope is a little smaller but this small change does not modify the discussion.

*What would be of interest here also is to compare this value with the measured alpha_0,exp, averaged for all 8 different pathlength measurements.*

Alpha_0, exp is deduced from the fitted slopes determined in Fig. 4b & 5b, so the value takes into account the full set of measurements at different pathlengths.

*L.254. Where does the line intensity come from? How was this estimated? There is a reference needed here. It also says "experimentally measured". By whom? In this work?*

The ground-state rotational lines of $N_2O$ in the millimeter-wave region are very well known. Line intensities are well determined thanks to a good knowledge of the permanent dipole moment of $N_2O$. A very good line profile analysis on the same transition shown in Fig. 5 with the same kind of THz source was performed in 2005 by F. Rohart et al.. This reference is now added to the manuscript.

*L337: "...by fitting.." what? A 'Voigt profile'? The function that is fitted to the data should be stated here ("...by fitting a Voigt profile..."). Moreover, in Figure 7 the absorbance spectrum of the R(22) of $N_2O$ line is shown. What do the authors mean by "integrated intensity" in Line 337?*

The rotational lines in Fig. 7 are fitted with a Voigt profile since at low pressure the Doppler and the collisional contributions of the broadening are rather close. The choice of the line profile is now stated in the manuscript. According to the remarks of reviewer 1, "integrated intensity" is replaced by "absorbance" L.337 and L.354.

*L338-342: It should be argued or shown that the "two baseline treatments" have no effect on the line shape and width. A comparison of results with and without the treatment could be shown here, since data manipulations like FFT filtering affect the line shape and hence the error of the resulting number density. A systematic error discussion could be included here.*

We have added a Fig. S5 in the Supplement describing the different steps of the post treatment used to reduce the oscillations due to standing waves occurring in MULTICHARME. It shows that only the rapid baseline oscillations are removed. The residual between raw and filtered data has an amplitude corresponding to around 5% of the total amplitude of the measured absorbance. With this figure, the reader can see that the line shape and the line width are preserved after the treatment. Now in the text we state that *"We have been careful that the post processing does not affect the line shape and hence the error of the resulting number density."*

*Figure 7: Red and blue circles (or panels) seem to have been mixed up. It would be meaningful to show the fit residuals in panel below the main figures. The unit of the integrated absorbance is stated in MHz, however as per the main text (L344) this should be wavenumbers. Please be uniform in your notation.*

We don't understand why reviewer1 says that the *Red and blue circles (or panels) seem to have been mixed up?* Nevertheless, according to the remark of reviewer1, we propose a new Fig.7 with the fit residuals below the main figures. The fitted area in the caption are logically in MHz units according to the X-axes units in Fig.7. The conversion MHz -> $cm^{-1}$ is required in a next step to determine the molecular density in molec/$cm^3$.

*Since the concentrations in the current experiment are significantly higher it is not clear how meaningful this comparison to the work by Itoh et al. is. Itoh et al. measured from a pressure of 6.7 mbar, which is not even as low as in the present study it seems? The conditions in the present paper are typical for chamber cleaning activities. Under these conditions it is known that the $O_3$ loss in the chamber is dominated by wall reactions and not by reactions with $O_2$. That is why these conditions are chosen to get rid of impurities, such as volatile organic compounds, on the chamber wall.*

***L406/407. Ozone being generated at atmospheric pressure? I thought the chamber was kept at low pressure in these experiments (see L375.)***

Itoh et al. have developed and experimentally verified a physical model allowing to understand the pressure and the wall material dependencies of the ozone-to-wall loss rate in a cylindrical tube (Itoh et al., 2004, 2011). In order to understand the differences between atmospheric pressure and low pressure measurements of the ozone lifetimes performed in CHARME, we thought it appropriate to mention the work of Itoh et al. even their low-pressure measurements are limited to few mbar. Effectively, the chamber was kept at low pressure during these experiments. Finally we converge to similar conclusions as RC1, so we have added at the end of section 3.2.3 the sentence proposed by RC1 in its report which allows to avoid any confusion: "*The pressure measurement conditions in THz rotational high-resolution spectroscopy are typical for chamber cleaning activities. Under these conditions it is known that the $O_3$ loss in the chamber is dominated by wall reactions and not by reactions with $O_2$. It corresponds to the first term of Eq. (1) and, therefore, as it is shown by our results, a decrease of the lifetime at low pressure was expected due to the reinforcement of the losses by the ozone diffusion on the walls chamber. That is why these conditions are chosen to get rid of impurities, such as VOCs, on the chamber wall.*"

***All the other remarks of RC1 (mainly typographical) have been taken into account in the new version of the manuscript. All the propositions of RC1 concerning the references have been taken into account. The references have been completed and verified***

---

## Author Comment (AC2)

**Response to RC2**

*The authors describe the successful characterisation and quantitative measurement capabilities of a new optical absorption instrument called MULTICHARME installed at the CHARME atmospheric simulation chamber in Dunkirk. The instrument is capable of measuring rovibrational transitions over the range of infrared to THz radiation with path lengths from 120 m to 280 m in the THz and up to 540 m in the IR range. Measurements of $N_2O$ and $O_3$ are shown, highlighting the potential for the distinction of isotopic composition and kinetic investigation.*

We would like to thank RC2 for its comments and time to read our work. Please find below the different responses, we have brought.

***Why was a zero-biased detector chosen for the THz radiation, instead of a typically more sensitive, powered alternative?***

There are presently no active sensors available in the THz region. On a laboratory scale it is possible to use a bolometer which is more sensitive than a zero-biased detector but requires cryogenic cooling. We observe significant standing wave that produce strong baseline variation in our measurements. Using a zero biased detector our measurements are limited by the baseline variations rather than the detector performance. We would therefore expect identical results using a bolometer.

***On Page 15, Line 370 you describe the loss processes for ozone before your THz measurement begins. However, if the losses occurred already during the ozone injection, the photometer should also have shown a lower value, no? And also, how long is the pumping time to reach the THz measuring pressure, such that it could explain the loss of half of the ozone? Is this consistent with the resulting wall losses in Section 3.2.3?***

Taking into account the remark of RC2, the sentence *"An unknown quantity of ozone was introduced at atmospheric pressure into CHARME. Based on the time calibration of the generator, performed with a photometric $O_3$ analyzer, and the injection time, the calculated number density is 500 ppmv. Then the ASC was pumped down to 1.5 mbar and the THz spectrometer was used to detect and quantify $O_3$ traces from individual rotational transitions"*

has been replaced by:

*"Based on the calibration of the ozone generator, performed with a photometric $O_3$ analyzer, and the injection time (90 min), the ozone volume ratio introduced in CHARME was estimated around 500 ppmv. Then the ASC was pumped down (in 45 min) to 1.5 mbar and the THz spectrometer was used to detect and quantify $O_3$ traces from individual rotational transitions. The $O_3$ wall losses occurring during the ozone introduction as well as during the pumping procedure contribute to reduce the ozone mixing ratio to an unknown lower value which will be measured by THz spectroscopy.".*

***Can you elaborate on the relatively larger error bars for the ozone detection between ~460 and 600 minutes in Fig. 8b, also with respect to how the LOD for ozone was determined?***

According to both remarks of the two reviewers, in the revised version, we detail the determination of the LOD at the end of the subsection 3.2.2. The batch process used in Fig.8b used the same method to determine the LOD which constitutes the error bars. We agree with reviewer 2 than the largest error bars are observed between 460 and 600 min. According to the observed absorbances, the measured S/N ratio is clearly weakened during this period. It seems that the phase of the standing waves, which affect the baseline, drift with the time. These oscillations could interfere destructively with the absorption signal leading to an irregularity in the profile's evolution and associated errors. We can assume that the dispersion variations, which occurs in parallel with the absorption variations, involves the phase drift of

the baseline oscillations. One more time, the only outcome is to find a way to correctly model the baseline variations due to standing waves in MULTICHARME. This work is under progress and will be the subject of a future publication.

***P16 L396: Why does the cleanliness of the chamber walls change with different ozone concentrations?***

The ozone wall losses are dependent on the cleanliness of the walls since adsorbed VOCs can react with ozone and thus contribute to reduce its lifetime in the gaseous phase. As example, Wang et al. (Wang et al., 2011) have found that overnight treatment with ozone mixing ratio in the range of 100 ppmv followed by a secondary vacuum pumping was suitable to reach much longer ozone lifetime in the CESAM chamber.

**All the technical corrections suggested by RC2 have been taken into account.**